# On the improvement of waves and storm surge hindcasts by downscaled atmospheric forcing: Application to historical storms

Émilie Bresson[1], Philippe Arbogast[1], Lotfi Aouf[2], Denis Paradis[2], Anna Kortcheva[3],
Andrey Bogatchev[3], Vasko Galabov[3], Marieta Dimitrova[3], Guillaume Morvan[2], Patrick Ohl[2],
Boryana Tsenova[3], and Florence Rabier[4]

[1]Centre National de Recherches Météorologiques - Groupe de Modélisation et d'Assimilation pour la Prévision, Toulouse, France
[2]Direction des Opérations pour la Prévision, Département Marine et Océanographie, Météo-France, Toulouse, France
[3]National Institute of Meteorology and Hydrology, Sofia, Bulgaria
[4]European Centre for Medium-Range Weather Forecasts, Reading, United Kingdom

*Correspondence to:* Émilie Bresson (emilie.bresson@gmail.com)

**Abstract.** Winds, waves and storm surges can inflict severe damage in coastal areas. In order to improve preparedness for such events, a better understanding of storm-induced coastal flooding episodes is necessary. To this end, this paper highlights the use of atmospheric downscaling techniques in order to improve waves and storm surge hindcasts. The downscaling techniques used here are based on existing European Centre for Medium-Range Weather Forecasts reanalyses (ERA-20C, ERA-40 and ERA-Interim). The results show that the 10 km-resolution data forcing provided by a downscaled atmospheric model gives a better waves and surges hindcast compared to using data directly from the reanalysis. Furthermore, the analysis of the most extreme mid-latitude cyclones indicates that a four-dimensional blending approach improves the whole process, as it assimilates more small-scale processes in the initial conditions. Our approach has been successfully applied to ERA-20C (the twentieth century reanalysis).

## 1 Introduction

One of the most vulnerable areas affected by winter storms are coastal regions, as their soils are often easily eroded and their population density is high (Barredo, 2007; Clarke and Rendell, 2009; Ferreira et al., 2009; Ciavola et al., 2011; André et al., 2013). Such storm events are frequently responsible for severe damages, significant economic losses and many casualties. In Europe, sensitive regions include the Atlantic, Mediterranean and Black Sea coasts; in particular, storm surges as high as 2.5 m have been recorded along the Atlantic coasts and 1.5 m along the western Black Sea coasts (Marcos et al., 2009; Ryabinin et al., 1996). These extreme events are often associated with winter low pressure systems; those that affect western Europe are principally mid-latitude cyclones that originate in the Atlantic ocean (Klawa and Ulbrich, 2003; Della-Marta et al., 2009; Usbeck et al., 2010), and the Bulgarian coasts are hit by cyclones generated in the Mediterranean region (Bocheva et al., 2007). The amplification of wind-generated waves and surge by equinox tides within deep low pressure systems can also produce a significant rise in sea level, resulting in coastal flooding.

For example, during the Xynthia storm, which hit the French Atlantic coast on February 27, 2010, a coastal flooding scenario occurred as a result of a tide coefficient of 102 that coincided with a highest astronomical tide between 0.96 m and 1.15 m and wind gusts of 160 km h$^{-1}$ over coastal regions and about 120 km h$^{-1}$ over land (Rivière et al., 2012). As a result of these conditions, a damaging storm surge crested above 1.60 m at La Rochelle and Les Sables d'Olonne. This example demonstrates that a better knowledge of the variability of these extreme coastal events is needed to improve high surf and storm surge warning systems. In addition, evaluating the frequency and severity of these events within the framework of ongoing climate change is equally critical. Consequently, a 20$^{th}$ century climatology of wave and storm surge would provide a useful baseline for coastal protection and risk management.

The lack of long-term wave records based on in-situ measurements and surge archives prevents the development of a completely observational 20$^{th}$-century climatology for waves and storm surges. Therefore, reconstructing wave and storm surge by hindcast using numerical models represents an alternative approach toward establishing a climatology. One straightforward method for hindcasting involves using global atmospheric reanalyses as the atmospheric forcing conditions in wave and storm surge numerical models (Reistad et al., 2011). Several weather forecast centers produce these global atmospheric reanalysis, including the European Centre for Medium-Range Weather Forecasts (ECMWF).

The ECMWF Re-Analyses (ERA) include different products that have various date ranges, spatial resolutions and assimilated datasets (Tab. 1;  Poli et al., 2013; Uppala et al., 2005; Dee et al., 2011). Although we can use the finer-scale reanalysis as initial conditions for a given period, a dynamical downscaling of the global reanalyses is also necessary, since they are too coarse to force the regional wave and storm surge models. Furthermore, certain mesoscale processes related to the formation of strong surface winds, such as sting jets (Hewson and Neu, 2015), are absent even in ERA-Interim, one of the higher-resolution reanalyses available from the ECMWF. Therefore, in order to better resolve mesoscale features associated with mid-latitude cyclone development and their interaction with locally-complex coastal topography, a dynamical downscaling can be applied on these reanalyses using a high resolution numerical model (e.g.,  Reistad et al., 2011; Li et al., 2016).

In this study, we apply two different downscaling methods on ERA datasets. The first one is a simple dynamical downscaling approach beyond the reanalysis truncation, whereas the second is more complex. We evaluate to what extent the mesoscale features resolved by the first downscaling technique impact our surge and wave reconstruction over the French and Bulgarian coasts, followed by an examination of the added-value of the second downscaling method against the first, simpler one. As observations are spatially and temporally scattered in these regions, we focus on thirty extreme events between 1924 and 2012 that targeted the French and Bulgarian coasts. The selected cases offer a large panel of observed extreme events with various affected areas (in particular, the French Atlantic and Mediterranean coasts and the Bulgarian Black Sea coast), including cases with more or less extended impacted zones, different cyclone trajectories and amplitudes and varied highest astronomical tide (Tab. 2). In the present paper, we first describe the methodology and data used for the downscaling strategies (Section 2.1) and then the wave and surge models configurations (Section 2.2). In Section 3, we first compare the results from the two downscaling techniques on reconstructing an intense cyclone's development, then we evaluate wave hindcasts and storm surge model skill, followed by an analysis of our early 20$^{th}$ century cases. Finally, Section 4 summarizes our conclusions.

## 2 Methodology

### 2.1 Dynamical downscaling of reanalyses

The general method of a dynamical downscaling uses a coarse resolution dataset, like global atmospheric reanalysis data, as initial conditions for a numerical atmospheric model. Three ECMWF reanalyses are selected for this study: ERA-20C,

ERA-40 and ERA-Interim (Tab. 1). They are all produced by older versions of the Integrated Forecasting System (IFS), the ECMWF's operational forecasting coupled model system. ERA-40 includes conventional observations (e.g. surface stations, buoys, radiosondes), polar satellites and geostationary satellites. ERA-Interim datasets benefit from improvements in assimilation methods and a large expansion of available data, with observation quantity and quality increasing over time. In order to mitigate this inhomogeneity in the $20^{th}$ century reanalysis, only observations of surface pressure and surface marine winds

are assimilated in the ERA-20C dataset. In order to provide the best possible atmospheric conditions for wave and storm surge hindcast, the following ERA datasets are downscaled for each event: ERA-20C for cases before 1957, ERA-40 for the 1957–1978 period, and ERA-Interim for storms occurring in 1979 and thereafter (Tab. 2). The designator "ERA-x" is used in this manuscript to describe a group of cases where more than one ERA reanalysis product is applied.

Hereafter, this study focuses on the advantages of downscaling global atmospheric reanalysis for the development of wave

and storm surge hindcasts. Over both the French and Bulgarian domains, numerical weather prediction (NWP) models require high horizontal and temporal resolution, especially for the storm surge model hindcast. For French events, the selected model, ARPEGE (Action de Recherche Petite Echelle Grande Echelle), is the operational global primitive-equation NWP system used at Météo-France and is based on the ARPEGE-IFS software developed in collaboration with ECMWF (Tab. 3; Courtier et al., 1991). A stretched grid allows for a finer horizontal resolution over France (around 10 km). The version used here has 70

hybrid vertical levels from 17 m to 70 km height. The Bulgarian events are hindcast from ALADIN (Aire Limitée, Adaptation dynamique, Développement InterNational) model, which is a limited-area model based on the ARPEGE system (Radnóti et al., 1995). The model's core characteristics are the same as for ARPEGE.

Two dynamical downscaling methods are examined here, hereafter referred to as D1 and D2, where D2 represents an improved version of D1. For D1, the necessary data from the global fields of ERA-x are interpolated to the plane model domain

both on horizontal and vertical scale for each NWP system, ARPEGE and ALADIN. The upper-air initialization step uses the spectral coefficients of ERA-x data. Then we apply the Schmidt transformation, which is well defined in spectral space to project the fields into the ARPEGE stretched grid. The land-surface initialization is not straightforward, since there are many differences between the ERA reanalysis and the NWP models in terms of the applied land-surface parameterizations and physiographic databases. For instance, the Tiled ECMWF Scheme for Surface Exchanges over Land (TESSEL) scheme of ERA-x

uses four soil layers with fixed thicknesses, each layer having its own water content. The land-surface scheme of ARPEGE, however, only uses two layers in our experiments; the top layer has a fixed size of 1 cm, and the second layer overlaps the first one and has a variable depth. Furthermore, for a given grid point, soil types are often very different in the two land-surface schemes. Therefore, using the raw land-surface datasets from ERA-x as initial conditions would be troublesome, since the water saturation fraction depends on the soil type. Thus, we interpolate the surface fields so as to preserve as much as possible

the ERA-x surface heat and momentum fluxes (Boisserie et al., 2016). The procedure is based on the conservation of the Soil Wetness Index (a relevant indicator for soil water availability) during the interpolation process, since soil water availability is supposed to regulate the partition of latent and sensible heat fluxes, which, in turn, influence energy and water exchanges between the atmosphere and the land-surface. The resulting files are initial conditions (IC-1) for the NWP forecasts (Fig. 1, top). Then, hourly forecasts are produced twice a day, at 00 UTC and at 12 UTC, starting from H+06 to H+18. The first six hours are not taken into account to prevent model spin up, and after H+18, the next forecast time is considered (Fig. 1, top). Forecasts are produced from a week (d-7) before to two days (d+2) after the day (d) that the storm impacted the coastline. The D2 method is more complex than D1 (Fig. 1, bottom). The D2 method also uses hourly forecasts produced twice a day, at 00 UTC and at 12 UTC, starting from H+06 to H+18, and the forecast start nine days (d-9) before and continue until two days after (d+2) the the day (d) that the storm impacted the coastline. Instead of using independent initial conditions (IC-1) like in D1 for the 00 UTC and 12 UTC forecasts, the initial conditions for D2 (IC-2) include information from the last 6-h forecast (Fig. 1, bottom). Consequently, the D2 method allows us to evaluate the importance of taking into account small wavelengths beyond the reanalysis truncation that are not considered in D1. Furthermore, after a short period of time (3 hours), non-linearities trigger small scale processes which are consistent with the large scale. This small-scale information provided by the 6-hour forecast is blended with the large scale information given by the interpolated reanalysis (IC-1) (Fig. 1, bottom). This procedure was cycled 4 times two days before the first 00 UTC forecast used as forcing for the wave and storm surge models. Therefore, the determination of one single initial condition (IC-2) uses 4 reanalyses. The D2 technique is applied to 10 recent French coastal flooding events (Tab. 2). These 10 cases represent a diverse panel of events affecting different coastlines with adequate observational data (satellite altimeters and tide gauges) to evaluate the reconstruction of the wave and storm surge observations and to enable a comparison between D1 and D2.

## 2.2 Description of wave and storm surge models

In order to ensure consistency in our case studies, the selected wave and storm surge models share similar general characteristics, despite being adapted specifically either the French or Bulgarian coasts.

### 2.2.1 Wave models

The French coast extreme wave events are hindcast with the Meteo-France WAve Model (MFWAM), a third-generation model of the operational wave forecasting system of Météo-France (Tab. 3). This model is based on the IFS-CY36R4 of the European wave model (ECWAM) with modified source terms for the dissipation by wave breaking and the air friction dedicated to swell damping as described in Ardhuin et al. (2010). The MFWAM model uses the wind input term as defined in Bidlot et al. (2005). The dissipation by wave breaking is directly related to the wave spectrum with a saturation rate of dissipation. The source term is a combination of an isotropic component and a direction-dependent component that controls the directional spread of the resulting wave spectra. It also includes a cumulative effect describing the smoothing of big breakers on small breakers. The term additionally uses a wave turbulence interaction component, which, as indicated in Ardhuin et al. (2010), is of secondary importance. The MFWAM model uses a quadruplet non-linear interaction term based on the discrete interactions

approximation as defined in the ECWAM model. In this study, a nested MFWAM model is implemented with a grid size of 0.1° for Western Europe, including the Mediterranean Sea. The domain boundaries are 20° N-72° N longitude, 32° W-42° E latitude (EURAT01 domain in Fig. 2). The wave spectrum is discretized in 24 directions and 30 frequencies starting from 0.035 to 0.58 Hz. This regional model is forced by boundary conditions provided by the global MFWAM model with a grid size of

0.5°. The global MFWAM model is driven by 6-hourly ERA-x winds. The SWAN (Simulating Waves Nearshore) model is used for the Bulgarian cases (Tab. 3). It is a third-generation wave model that is especially designed to simulate waves in near-shore waters and is often applied to enclosed and semi-enclosed seas, estuaries and lakes (Booij et al., 1999). The model computes random, short-crested, wind-generated waves in coastal regions and inland waters. SWAN accounts for wave propagation and transitions from deep to shallow water at finite depths by solving the spectral wave action balance equation, which includes

source terms for the wind input, non-linear interactions, whitecapping, bottom friction and depth-induced breaking. The model performance, the parameterizations of the wave generation and dissipation processes and other aspects of SWAN applied to the Black Sea basin have been addressed in previous studies (Akpinar et al., 2012; Arkhipkin et al., 2014; Rusu et al., 2014). The model domain that is used for the simulations of our historical Black Sea storms is based on a numerical grid covering the entire Black Sea area (40° N-47° N and 27° E-42° E; hereafter named BUL; Fig. 2) with a mesh size of 0.0333° in latitude

and longitude. The spectral discretization is based on 36 directions and 30 frequencies logarithmically spaced from 0.05 Hz to 1.00 Hz. The wind input parameterization follows Komen et al. (1984), and whitecapping is based on Hasselmann (1974), with the $\delta$ coefficient (which determines the dependency of whitecapping on wave number) set to 1 (following Rogers et al., 2003). This specific set of parameterizations is chosen to have the lowest bias, root mean square error (RMSE) and scatter index when compared to results from the model and the along-track satellite altimetry data. The bathymetry data for the wave model are

obtained by the digitalization of proprietary maps provided by the Bulgarian military hydrographic service.

### 2.2.2  Storm surge models

The operational surge model of Météo-France (Daniel et al., 2001) is a barotropic 2-dimensional version of the HYCOM model (HYbrid Coordinate Ocean Model) implemented by SHOM (Service Hydrographique et Océanographique de la Marine) from the 3-dimensional version (Tab. 3; Bleck, 2002; Baraille and Filatoff, 1995). The HYCOM code is managed by an international

consortium, including COAPS (Center for Ocean-Atmospheric Prediction Studies, USA), NRL (Naval Research Laboratory, USA), SHOM (France), DMI (Danish Meteorological Institute, Denmark) and NERSC (Nansen Environmental and Remote Sensing Center, Norway). The model is run on two domains (as shown in Fig. 2); ATL corresponds to the North-East Atlantic area (Bay of Biscay, English Channel and North Sea) from 43° N to 62° N and from 9° W to 10° E, and MED defines the Mediterranean Sea domain from 30° N to 46° N and from 9° W to 37° E. In both domains, the model runs on a grid

size of approximately 1 km on the French coast (curvilinear grid). The tides imposed at the marine boundaries are computed according to the 17 harmonic components from the COMAPI (COastal Modelling for Altimetry Product Improvement project) regional atlas implemented in the North East Atlantic Ocean area (Cancet et al., 2010). The bottom friction coefficient is spatially variable and has been optimized to properly reproduce the propagation of tides. Tides are discarded in the storm surge computation, for which another computation of the tides, based in harmonic components obtained from measurements by

SHOM, is added to the storm surge in order to more accurately represent the sea level at specific locations. The bottom friction coefficient is constant and taken as equal to 0.002. For both HYCOM configurations (ATL and MED), the drag coefficient used to compute the wind stress follows the Charnock (1955) scheme with a constant Charnock parameter of 0.025.

The simulations of storm surges for Black Sea cases are based on the storm surge model of Météo-France (Daniel et al., 2001), which was adapted for the Black Sea in Mungov and Daniel (2000) (Tab. 3). The model is depth-integrated, and tides are not taken into account, as their amplitude is less than 9 cm in the Black Sea. The model grid for the Black Sea is a regular spherical grid with a spatial resolution of $0.0333°$ that covers the entire Black Sea. The bottom friction coefficient is $1.5 \ 10^{-3}$ over the shelf. In addition, the depth of the Black Sea mixed layer is considered as a liquid bottom given the very stable stratification of the Black Sea waters and the shallowness of the mixed layer depth, and as such, the bottom friction coefficient is defined as $1.5 \ 10^{-5}$ over the liquid bottom. Data about the seasonal variations of the Black Sea mixed layer depth are taken from the study by Kara et al. (2009). Without this liquid bottom setup, the depth-integrated models for the Black Sea fail to simulate any surge, even if strong, constant winds are used as input. The bathymetry data for the storm surge model were obtained by digitizing proprietary maps provided by the Bulgarian military hydrographic service.

## 3 Results

### 3.1 Impact of the two downscaling techniques on a deep cyclone development

The effects of the two downscaling techniques on the reconstruction of intense storms are presented for the case of the Lothar storm, an extreme cyclogenesis event (occurring a few hours before the Martin storm described further in Sections 3.2 and 3.3) in December 1999. It is the most severe storm in terms of pressure gradient, surface winds and displacement velocity to hit France within the observational record (Wernli et al., 2002; Rivière et al., 2010). This storm did not produce extreme wave and storm surge, and thus it was not selected for hindcasts. Nevertheless, it is interesting to look at the behaviour of both downscaling strategies for this particular case due to its uniquely tight horizontal pressure gradient. For this storm, the D1 method slightly improves the ERA-Interim reanalysis fields, but the D2 downscaling better reproduces the cyclone structure over Northern France (Fig. 3). A statistical analysis using the mean, the bias, the root mean square error (RMSE) and the standard deviation error (STD) is performed with the 12 meteorological stations available in an area encompassing the low pressure system (48° N-50° N; 2° E-4° E). This analysis confirms that the use of D1 forcing is an improvement compared to using an ERA-Interim reanalysis with respect to surface observations. The use of D2 slightly improves the reconstruction of the observations ((Table 4)).

### 3.2 Wave hindcasts

For the wave reconstruction evaluation, simulated Significant Wave Heights (SWH) are compared against observations from satellite altimeter data and in-situ observations. Several satellites operated over the French and Bulgarian coasts during the storms: TOPEX-Poseidon (1992–2005), ERS2 (1995–2011), ENVISAT (2002–2012) and Jason-1 (2002–2013). In addition,

buoys and Acoustic Doppler Current Profiler (ADCP) in-situ provide SWH information. The limited scope of each of these observational datasets, together with the coarse resolution of altimeter measurements, preclude a comprehensive validation for all the selected cases. For an initial evaluation of our modelling approach, the results from the wave model driven by ERA-x and D1 data is compared to available altimeter data. The simulated wave heights are collocated with the altimeter tracks within a time window of 3 hours. For the 2004, 2007, 2008 and 2010 French Atlantic coast storms and the 2012 Bulgarian storm, data are collected from two satellite altimeters, Jason-1 and ENVISAT. The scatter plots between model and altimeter wave heights indicate that the use of D1 winds provides a better fit to the data (Fig. 4). In particular, when compared to the results for the wave model driven by ERA-Interim initial conditions, the use of D1 data reduces the normalized root mean square error (NRMSE) from 17.1 to 13.1 %, largely owing to a significant reduction of bias from -35 to -4 cm (Fig. 4). The D1 downscaling also leads to a better fit for high SWH, providing an important validation for extreme wave events. For the 1998, 1999 and 2000 storms, altimeters wave heights from TOPEX and ERS2 are also used for the evaluation of the modelled SWH, and the same tendency is found, with an improvement of the reconstruction of SWH using D1 winds over ERA-x winds (not shown).

As satellite altimeters provide data along a track, these observations can be useful for mapping the spatial distribution of the SWH. For further examination, we present the 2012 Bulgarian storm as an example of a more detailed evaluation of the reconstruction against observations. The wave model outputs using ERA-Interim or D1 initial conditions are first compared to the 214 along-track data points measured by the Jason-1 and ENVISAT satellite altimeters on 7 and 8 February, 2012. The wave reconstruction given by D1 forcing more closely matches the satellite observations, especially in terms of wave intensity over the southern part of the satellite track (Fig. 5). However, the maximum observed SWH value is not reached by the model for both the ERA-Interim winds and the D1 winds. Regarding the temporal evolution of the 2012 Bulgarian storm, we can use in-situ acoustic Doppler current profiler (ADCP) to check if the peak SWH occur at the same time in the observations and the reconstruction. In Fig. 6, we compare the SWH data from the ADCP located at Pasha Dere beach at 20 m depth provided by the Bulgarian Institute of Oceanology (Valchev et al., 2014) to our wave model outputs. The use of D1 generally overestimates the measured SWH, while the use of ERA-Interim underestimates the wave heights. However, the use of D1 winds leads to a better matching of the temporal structure of the wave. The overall improvement of the SWH reconstruction by using D1 is confirmed by the statistical analysis in Table 5. The temporal evolution of a storm can also be evaluated with in-situ buoys. For example, for the 2010 Mediterranean storm, we compare the time series of SWH from model and buoy data (43.4° N and 7.8° E) off the coast of Nice, France, at the peak of the storm (Fig. 7). The results show that the SWH induced by using D1 data more closely match the buoy observations when compared to the ERA-Interim data forcing. Given our validation of the D2 approach discussed in Section 3.1, the D2-driven SWH hindcast of the 2004, 2007, 2008 and 2010 French Atlantic storms are also compared to satellite altimeter data. The statistical analysis (bias and NRMSE) reveals that the use of D2 winds leads to better results than the use of D1 winds (Fig. 8). Biases of SWH are slightly improved using D2 winds over D1 winds; however, D2 winds slightly increase the NRMSE of SWH for the 2004, 2007 and 2008 storms. The D2 method only slightly improves the NRMSE of SWH for the storm Xynthia (February 2010). While the application of the D2-method winds does not lead to an improved result over D1 in all cases, D2 appears to show better skill for events with higher wind speeds, such as the ones observed during the Lothar storm.

### 3.3 Storm surge hindcasts

Storm surge hindcasts can be evaluated by tide gauge measurements. A network of 25 tide gauges along the French coasts is maintained to validate the surge model implemented at Météo-France. Furthermore, an additional 12 hydro-meteorological stations are located along the Bulgarian coasts for validation purposes. Depending on the storm extent and instrument condition, the number of available data points is different for each storm (Tab. 2). For a global evaluation of hindcast regarding tide gauges, all the available measurements with a peak in storm surge are selected. A Weighted Normalized Observation Error ($WNOE$) is calculated to highlight the overestimation and underestimation of the simulated maximum storm surges with respect to available measurements, and it is defined by the following equation:

$$WNOE = 100.\alpha(t_{sim}).\left(\frac{X_{sim} - X_{mea}}{X_{mea}}\right)$$

In this simple calculation, $t_{sim(mea)}$ is the time related to the simulation outputs (measurements) (in hours), $X_{sim(mea)}$ is the simulated (measured) value of maximum storm surge (in cm), and $\alpha$ is the weighting coefficient. The value of $\alpha$ is equal to 0.9 if the simulated maximum of storm surge falls within a time window of +/- 3 h with respect to the observed peak time; if it is sooner or later, the weighting coefficient is set equal to 1.1 to reflect greater bias. For some cases, when no time information is available, no weighting is applied, and thus $\alpha = 1$. When $\|WNOE\| < 20\%$, we consider errors to be low or moderate. Moreover, the values are evaluated regarding the number of samples (Tab. 6). First, we evaluate the impact of using wind and mean sea level pressure data from D1 instead of from ERA-x. The storm surge outputs using ERA-x forcing have a tendency to underestimate maximum storm surge compared to D1 forcing (Fig. 9, 10 and 11). Cases with low or moderate errors represent a larger proportion of storm surge events when D1 data are used. In particular, 63% of storm surge events were associated with low and moderate error in the ATL basin, 54% for BUL and 100% for the MED domain. This represents a general improvement over the ERA-x data, which had low/moderate errors for 21% of storm surge events for ATL, 0% for BUL and 100% for the MED domain (Tab. 7).

Second, the D2 method is applied on two examples of storm surge reconstruction (the Atlantic 2004 and 2007 storms in France) with a corresponding statistical analysis. For the December 2004 storm, a deep low of 980 hPa crossed the northern French coasts from west to east, generating high waves and surge along the British Channel and the North Sea coasts due to strong northwesterly winds wrapping behind the system. The maximum observed surge exceeded 1 m at St Malo and Dunkirk during a period of below-average tide (Fig. 9). Over the course of this event, the application of ERA-Interim winds result in an underestimation of the surge by roughly 60 cm at St Malo and 20 cm at Dunkirk (Fig. 9). However, the use of D1 forcing successfully captures the peak of the surge in St Malo and Dunkirk. The use of D2 winds induces an overestimation of the surge of 20 cm at St Malo and roughly the same surge as D1 at Dunkirk. The second example of storm surge hindcast is provided by the November 2007 storm. This event affected the whole North Sea (including Dunkirk and Calais on the French coast) and parts of the eastern British channel. It was associated with a strong northwesterly wind on the North Sea and lasted nearly 24 hours. At the peak of the storm event, a surge of 2.30 m was recorded at Dunkirk (Fig. 10). While the ERA-x forcing significantly underestimates the surge by 80 cm (Fig. 10), a good fit is obtained by the model with both the D1 and D2 data forcing. For this particular storm, the D2 winds give slightly better surge results on 11 November, 2007, at 00 UTC. These two

storms are examples of the various responses of the storm surge hindcast with both types of downscaling: no significant trend could be highlighted. Overall, the dispersion of WNOE values for the D2 results is larger than for D1 (Fig. 11), and Atlantic cases are better hindcasted with D2 forcing data (Table 7). The ability of D2 to simulate very deep cyclones could explain this point, since the mesoscale processes involved in strong wind are better described with the D2 approach.

## 3.4 Evaluation of early 20$^{th}$ century cases hindcast using ERA-20C

The 20$^{th}$ century extreme events that occurred before 1957 can be hindcast by using ERA-20C, the 20$^{th}$ century reanalysis ECMWF project (Poli et al., 2013). For these cases, even if there were no available wave observations, a storm surge evaluation is possible due to the availability of reliable sea-level observations.

To validate the concept of downscaling using ERA-20C reanalyses, we concentrated on the major storm that occurred in the North Sea in February, 1953 (Fig. 12). It caused severe damage along the Dutch, Belgian and English coasts. Wind intensity around force 10 on the Beaufort scale (around 90 km h$^{-1}$) were measured in Scotland and Northern England. The winds and the low atmospheric pressure combined with exceptional equinox tides were responsible for the surge, which was exacerbated as well by the funnel shape and shallowness of the North Sea. The Netherlands were the worst-affected, recording 1,836 deaths and widespread property damage (Gerritsen, 2005). Most of the casualties occurred in the southern province of Zeeland; an additional 307 people were reported killed in England, 19 in Scotland and 28 in Belgium as a result of the storm. The most striking feature along the Dutch coast was a long swell with a peak period of 20 s, which induced wave flooding. In our reconstruction of the event, the MFWAM results using the D1 winds indicate SWH exceeding 16 m in the western part of the North Sea at 00 UTC on 1 February, 1953 (Fig. 13). The storm surge hindcast produces a high surge which is unusual for this area; in particular, along the Dutch and Belgian coastlines storm surges exceeded 3 m either with ERA-20C or D1 data forcing (Fig. 14). The improvement of storm surge reconstruction induced by D1 forcing was particularly marked at Ijmuiden, Ostend, Brouwershavn and Dieppe, where the recorded peaks of storm surge are better represented than for ERA-20C (Fig. 15).

## 4 Conclusions

ECMWF reanalyses data are widely used for many climatological studies. However, due to the coarse spatial resolution and the limited temporal resolution of reanalysis model output, there is significant bias for high wind speeds associated with extreme midlatitude cyclones. To overcome this problem, dynamical downscaling techniques are implemented and applied to reproduce high resolution historical atmospheric fields. ERA-20C, ERA-40 and ERA-Interim data are used to encompass the studied period of 1924 – 2012. Very short range forecasts using 10 km resolution and hydrostatic models initialized with ERA-x analyses provide the downscaled data, which are used in turn to force wave and storm surge numerical models. This approach was already tested for the North Sea coast for a long period using only ERA-40 data. In order to evaluate such downscaling technique on different initial conditions, thirty cases are selected over French and Bulgarian coastlines to offer a diverse selection of storm characteristics in terms of location, intensity, highest astronomic tide and meteorological context. Some early 20$^{th}$ century cases generating extreme storm surge and waves are part of this selection due to the recent availability

of ERA-20C. This study shows a significant and quasi-systematic improvement of wave and storm surge hindcast when using downscaled winds. The evaluation with independent wave observations (such as wave heights from altimeters) shows the strong reduction of bias and improved RMSE of significant wave height for extreme waves events. The downscaling techniques are also well-suited for storm surge extreme events, such as the 1953 storm, since the storm surge reconstruction using the presented approach fits with the recorded data from the Belgian and Dutch coasts. The D2 method, generally leads to an improvement in comparison with D1, especially for cases with small-scale, intense mid-latitude cyclones. Dynamical downscaling is a promising technique for providing an accurate reconstruction of waves and storm surges for the $20^{th}$ century. After evaluation and calibration with observations, these model outputs can be useful to analyze the interannual variability of coastal wind-storms and to improve the thresholds used in the wave submersion warning system. Regional climate modelling in future studies is expected to address the response of wave and surge extreme variability to storm-track modifications due to global climate change. A further step towards this objective would be to use interactive models of wave and storm surge to enhance the hindcast. We expect that these approaches to reconstructing extreme events will prove valuable for coastal protection and risk management.

*Acknowledgements.* The research was carried out as part of the IncREO (Increasing Resilience through Earth Observation) project with funding from the European Union Seventh Framework Programme under the grant agreement n°312461. The authors would like to thank the European Commission for its financial support through the 7th framework programme project. We are also most grateful to Françoise Taillefer for her unconditional technical support and François Bouyssel for his constructive and valuable advice about the second downscaling method. Special thanks go to Philippe Dandin for his involvement in setting up this project. We also thank SHOM for providing the French storm surge measurements. Christophe-Thomas Simmons is warmly thanked for helping to improve the manuscript.

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

**Table 1.** Characteristics of ERA-20C, ERA-40 and ERA-Interim reanalyses. 4(3)D-Var: 4(3)-dimensional variational analysis; VarBC: Variational Bias Correction of surface pressure observations.

|  | ERA-20C | ERA-40 | ERA-Interim |
|---|---|---|---|
| Time period | 1900 – 2010 | 1957 – 2002 | 1979 – present |
| IFS version | Cy38r1 | Cy23r4 | Cy31r2 |
| Data assimilation system | 24-hour 4D-Var; VarBC | 6-hour 3D-Var | 12-hour 4D-Var; VarBC |
| Spectral resolution | T159 ($\sim$ 125 km) | T159 ($\sim$ 125 km) | T255 ($\sim$ 80 km) |
| Number of vertical levels | 91 | 60 | 60 |
| Vertical scale (from the surface up to) | 0.01 hPa ($\sim$ 80 km) | 0.1 hPa ($\sim$ 64 km) | 0.1 hPa ($\sim$ 64 km) |
| Pressure levels | 37 | 23 | 37 |
| Reference | Poli et al. (2013) | Uppala et al. (2005) | Dee et al. (2011) |

**Table 2.** List of the 30 cases selected for this study. Coast: Atl. - Med. for Atlantic and Mediterranean. Tide gauges: number of available and useful tide gauges. Storm surge (m): maximum storm surge recorded. Star is for unknown information.

| Coast | Date | Tide gauges | Storm surge | Downscaling | ECMWF reanalyses |
|---|---|---|---|---|---|
| Atlantic | 8 Oct. 1924 | * | * | D1 | ERA-20C |
| | 14 Mar. 1937 | * | * | D1 | ERA-20C |
| | 31 Jan - 1 Feb. 1953 | * | > 3 | D1 | ERA-20C |
| | 13 Feb. 1972 | 10 | 1.83 | D1 | ERA-40 |
| | 30 Nov. - 2 Dec. 1976 | 12 | 1.36 | D1 | ERA-40 |
| | 11 - 13 Jan. 1978 | 7 | 1.65 | D1 | ERA-40 |
| | 15 - 16 Oct. 1987 | 12 | 1.72 | D1 | ERA-Interim |
| | 26 Feb. - 1 Mar. 1990 | 6 | 1.67 | D1 | ERA-Interim |
| | 2 - 4 Jan. 1998 | 5 | 1.60 | D1 / D2 | ERA-Interim |
| | 6 Nov. 2000 | 8 | 1.00 | D1 / D2 | ERA-Interim |
| | 17 Dec. 2004 | 7 | 1.30 | D1 / D2 | ERA-Interim |
| | 9 Nov. 2007 | 2 | 2.20 | D1 / D2 | ERA-Interim |
| | 10 Mar. 2008 (Johanna) | 7 | 1.30 | D1 / D2 | ERA-Interim |
| | 23 - 24 Jan. 2009 (Klaus) | 10 | 1.29 | D1 | ERA-Interim |
| | 28 Feb. 2010 (Xynthia) | 8 | > 1.60 | D1 / D2 | ERA-Interim |
| Mediterranean | 6 Nov. 1982 | * | * | D1 | ERA-Interim |
| | 6 - 7 Feb. 2009 | 7 | 0.60 | D1 / D2 | ERA-Interim |
| | 24 - 25 Dec. 2009 | 6 | 0.50 | D1 / D2 | ERA-Interim |
| | 19 Feb. 2010 | 6 | 0.50 | D1 / D2 | ERA-Interim |
| Atl. - Med. | 27 Dec. 1999 (Martin) | 4 | 1.60 | D1 / D2 | ERA-Interim |
| Bulgarian | 5 - 21 Oct 1976 | 2 | 1.00 | D1 | ERA-40 |
| | 16 - 21 Jan. 1977 | 1 | 0.60 | D1 | ERA-40 |
| | 13 - 23 Feb. 1979 | 3 | 1.43 | D1 | ERA-Interim |
| | 7 - 10 Jan. 1981 | 0 | * | D1 | ERA-Interim |
| | 24 - 31 Dec. 1996 | 2 | 1.00 | D1 | ERA-Interim |
| | 15 - 19 Dec. 1997 | 1 | 1.30 | D1 | ERA-Interim |
| | 20 - 27 Jan. 1998 | 2 | 0.90 | D1 | ERA-Interim |
| | 1 - 3 Jul. 2006 | 2 | 0.60 | D1 | ERA-Interim |
| | 8 - 11 Mar. 2010 | 2 | 0.90 - 1.00 | D1 | ERA-Interim |
| | 7 - 9 Feb. 2012 | 2 | * | D1 | ERA-Interim |

**Table 3.** Outline of the numerical models required for wave and storm surge hindcasts.

| Purpose | Model | Resolution | Coupling - Initial conditions data | Domain |
|---------|-------|-----------|-----------------------------------|--------|
| Atmosphere | ARPEGE D1 | T798 ($\sim 10$ km) | ERA-x | global |
| | ARPEGE D2 | T798 ($\sim 10$ km) | ERA-x + ARPEGE | global |
| | ALADIN | 10km | ARPEGE D1 | Bulgaria |
| Wave | MFWAM | $0.1°$ | ARPEGE D1/D2 | Western Europe |
| | SWAN | $0.1°$ | ALADIN | Bulgaria |
| Surge | HYCOM | 1 km | ARPEGE D1/D2 + bathymetry | ATL |
| | HYCOM | 1 km | ARPEGE D1/D2 + bathymetry | MED |
| | MF model | $0.0333°$ | ALADIN + bathymetry | Black Sea |

**Table 4.** Statics for MSLP from ERA-Interim reanalysis at 06 UTC 26 December 1999, 12-h forecast using the D1 and D2 at 18 UTC 25 December 1999, versus observations at 06 UTC 26 December 1999. Mean (hPa), standard deviation error (STD; hPa), bias (hPa), Root Mean Square Error (RMSE; hPa). Calculations are done for the nearest point. Small domain corresponds to $48°$ N-$50°$ N; $2°$ E-$4°$ E and includes 12 pairs of data and model values.

| | Mean | STD | Bias | RMSE |
|---|------|-----|------|------|
| Obs | 973 | 2 | – | – |
| ERA-Interim | 993 | 10 | 12 | 18 |
| D1 | 980 | 1 | 6 | 6 |
| D2 | 977 | 1 | 5 | 5 |

**Table 5.** Comparison of SWAN wave model SWH (m) and altimeter data from ENVISAT and Jason-1 satellites for the 2012 case over the Bulgarian coast.

| Time of satellite track | | Pairs | Mean | | | Biais | | RMSE | | Scatter Index | |
|-------------------------|---|-------|------|-------------|-----|-------------|-------|-------------|------|-------------|------|
| | | | Obs | ERA-Interim | D1 | ERA-Interim | D1 | ERA-Interim | D1 | ERA-Interim | D1 |
| 7 Feb. 2012 | 08 UTC | 44 | 3.9 | 3.5 | 4.1 | -0.43 | 0.21 | 0.60 | 0.37 | 0.15 | 0.10 |
| | 14 UTC | 76 | 3.6 | 3.2 | 3.8 | -0.41 | 0.15 | 0.66 | 0.57 | 0.18 | 0.16 |
| | 20 UTC | 51 | 6.4 | 5.3 | 6.3 | -1.08 | -0.09 | 1.14 | 0.37 | 0.18 | 0.06 |
| 8 Feb. 2012 | 14 UTC | 43 | 5.6 | 4.4 | 4.7 | -1.22 | -0.94 | 1.37 | 1.16 | 0.24 | 0.21 |

**Table 6.** Number of observations used for calculations of $WNOE$ for each region and each forcing.

|  | ERA-x | D1 | D2 |
|---|---|---|---|
| ATL | 34 | 34 | 15 |
| MED | 13 | 13 | 13 |
| BUL | 9 | 9 | 0 |

**Table 7.** Portion of cases (%) with $\|WNOE\| < 20\%$ for each coast (ATL: Atlantic; MED: Mediterranean Sea; BUL: Bulgarian; common cases: cases using D1 and D2 forcing).

|  | ERA-x | D1 | D2 |
|---|---|---|---|
| ATL | 21 | 63 | 80 |
| MED | 0 | 54 | 38 |
| BUL | 33 | 100 | – |
| Common cases | 18 | 64 | 61 |

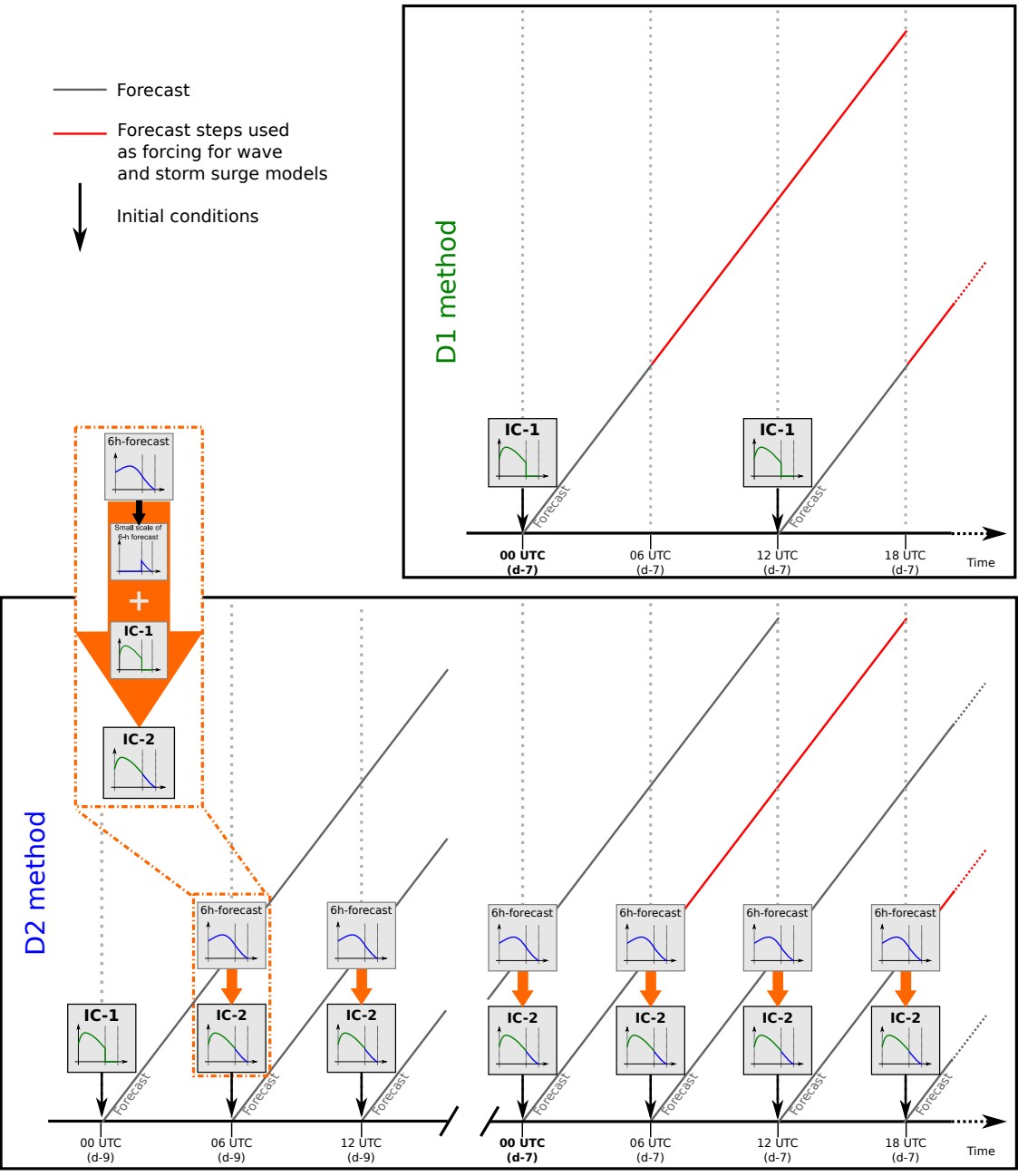

**Figure 1.** Schematic representation of D1 and D2 techniques. Energy spectra are within small stamps. The red part of forecast are the forecast data used as input forcing in the wave and storm surge models.

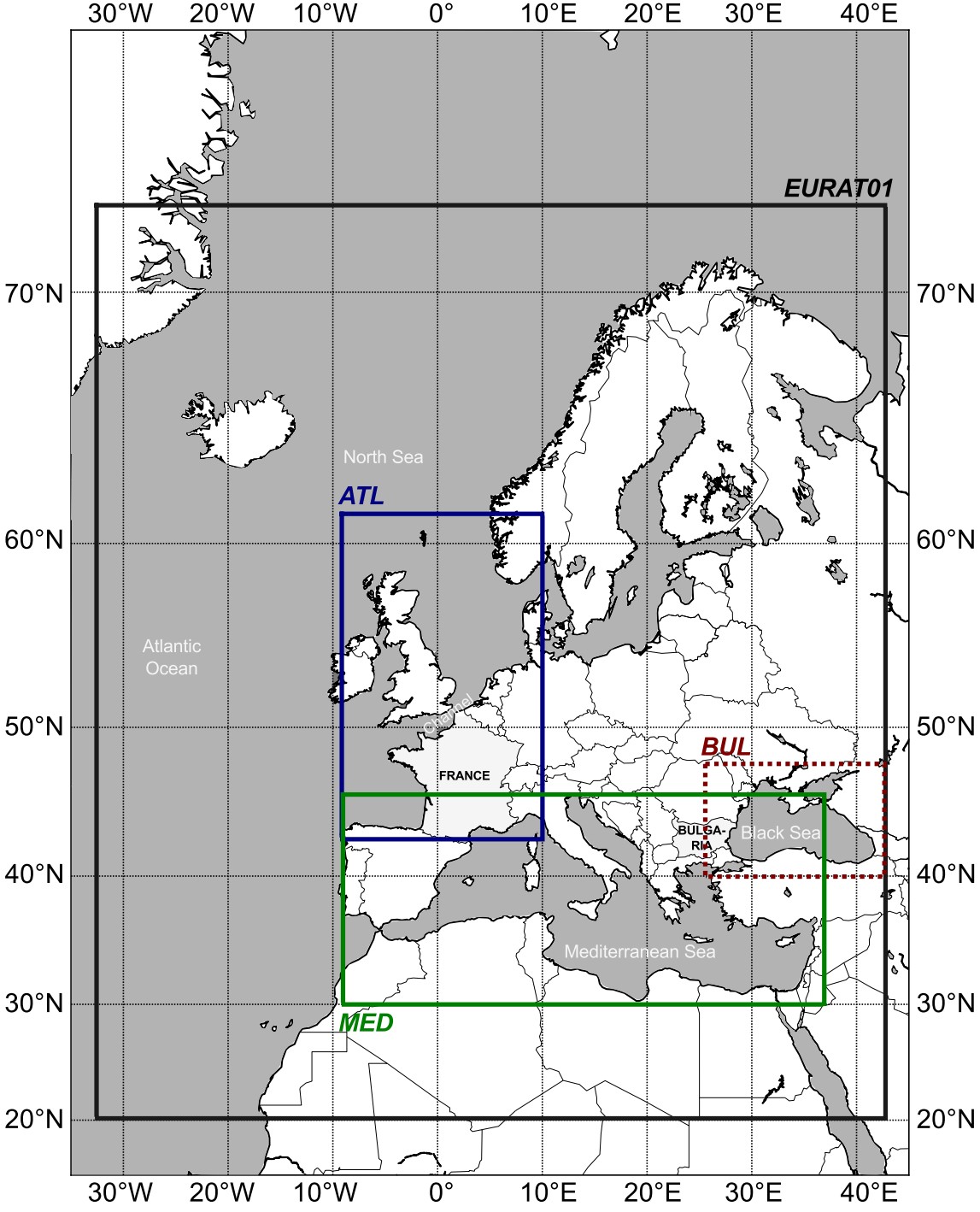

**Figure 2.** Locations of EURAT01 (black), ATL (blue), MED (green) and BUL (red) domains used in the study, respectively for European 0.1° resolution grid and Atlantic, Mediterranean and Bulgarian domains.

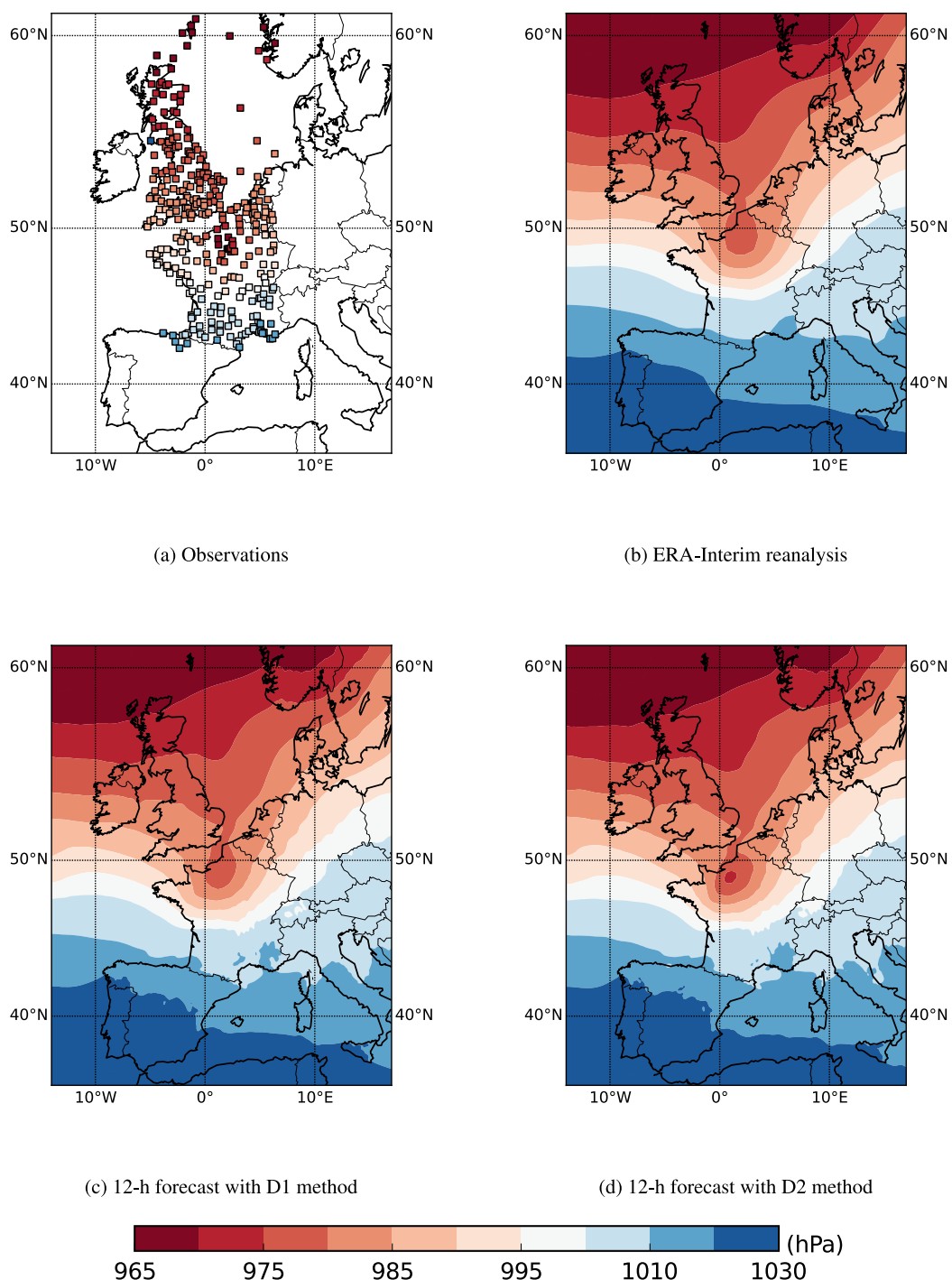

(a) Observations

(b) ERA-Interim reanalysis

(c) 12-h forecast with D1 method

(d) 12-h forecast with D2 method

965    975    985    995    1010    1030    (hPa)

**Figure 3.** Mean-sea level pressure (hPa) from observations (a) and ERA-Interim reanalysis at 06 UTC 26 December 1999 (b), from 12-h forecast using the D1 (c) and D2 (d) downscaling methods at 18 UTC 25 December 1999.

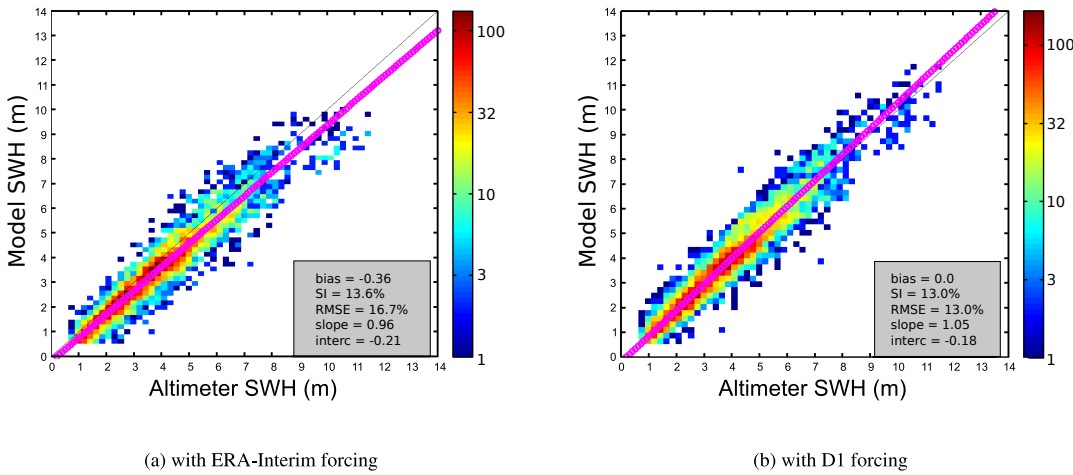

(a) with ERA-Interim forcing
(b) with D1 forcing

**Figure 4.** Scatter plots of significant wave heights (SWH) of model MFWAM and altimeters (ENVISAT and Jason-1) for the 2004, 2007, 2008 and 2010 French storms. (a) and (b) stand for runs with interpolated ERA-interim and D1 wind forcing, respectively.

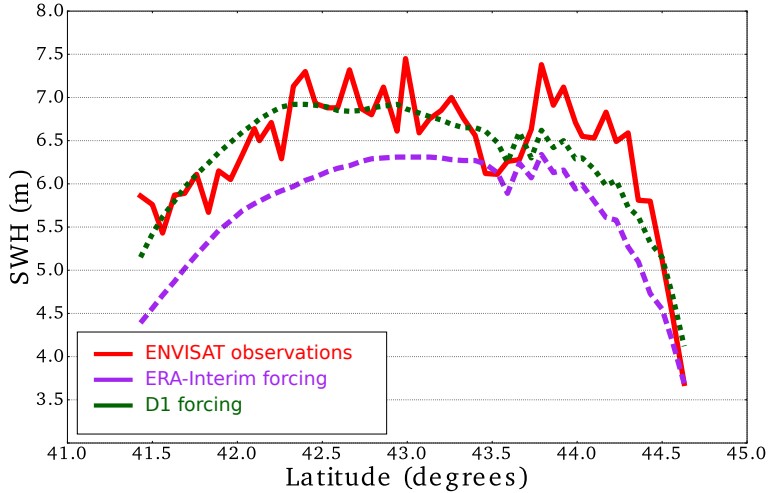

**Figure 5.** Comparison of the simulated significant wave heights (SWH) with downscaled wind input and ERA-Interim wind input with the data from the ENVISAT track crossing the Western Black Sea at 20 UTC on 7 February, 2012. Purple and green colors stand for ERA-Interim and D1 forcing, respectively. Red line stands for ENVISAT observations.

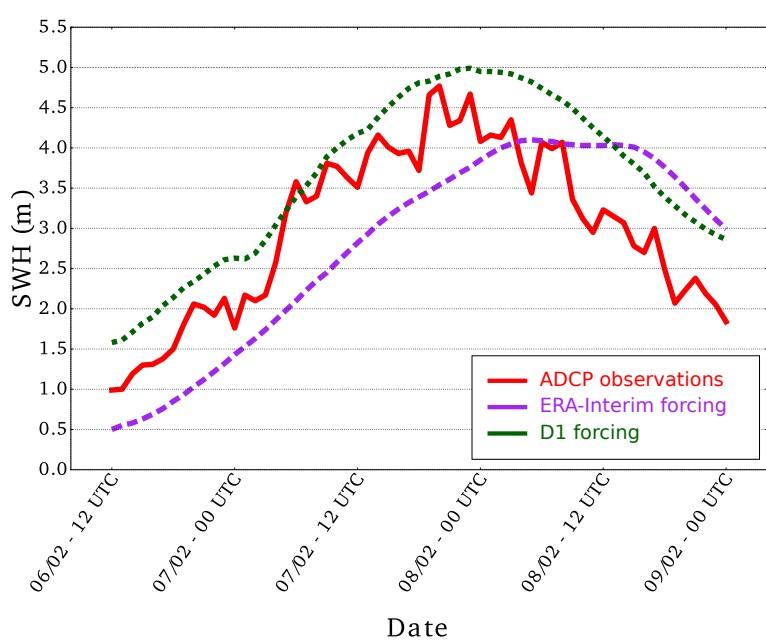

**Figure 6.** Comparison of the simulated significant wave heights using the two wind inputs (downscaled wind input D1 and ERA-Interim) with the data by ADCP located on the western Black Sea coast at 20 m depth during the storm of 7-8 February 2012. ADCP location coordinates: 43°04'49" N - 28°01'40" E. Purple and green colors stand for ERA-Interim and D1 forcing, respectively. The red line represents the ADCP observations.

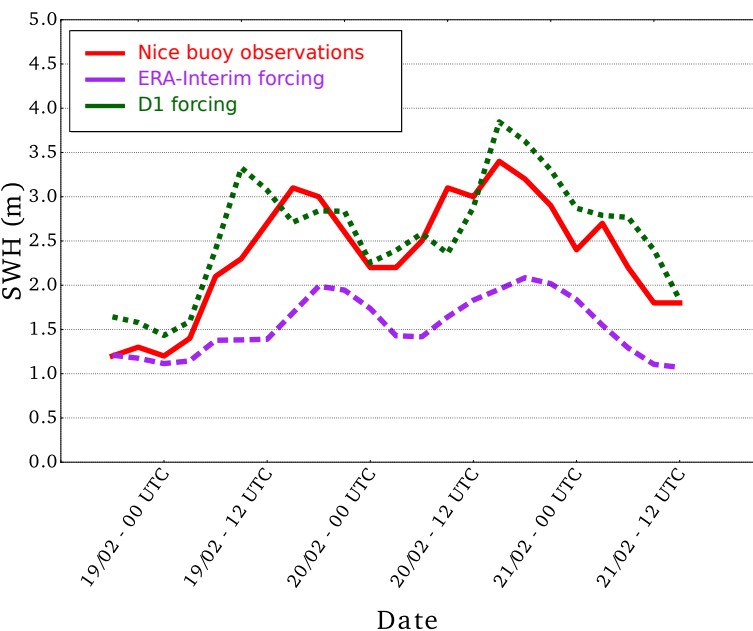

**Figure 7.** Time series of significant wave heights (SWH) for the storm on February 2010 near Nice (43°24'0" N - 7°48'0" E) in the Mediterranean Sea. Purple and green colors stand for ERA-Interim and D1 forcing, respectively. The red line shows the time series of the Nice buoy observations.

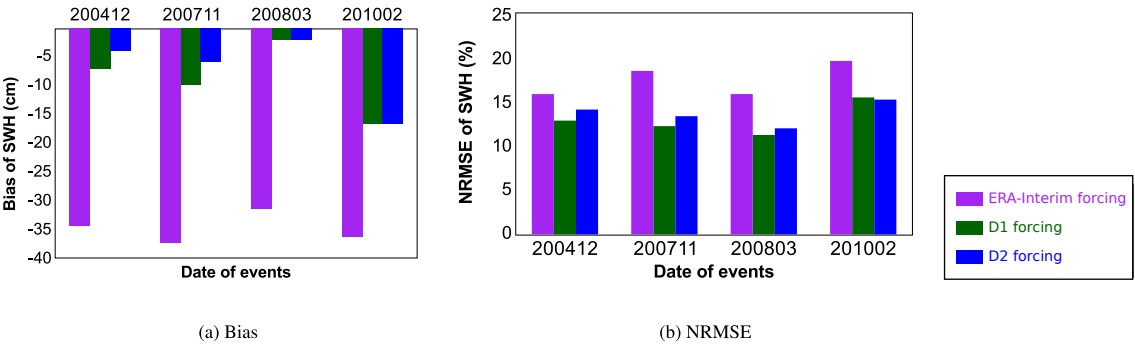

(a) Bias

(b) NRMSE

**Figure 8.** Variation of the bias (a) and the normalized root mean square error (NRMSE; b) of significant wave heights from the model MFWAM in comparison with the altimeters (ENVISAT and Jason-1) for the 2004, 2007, 2008 and 2010 French storms. Purple, green and blue colors stand for ERA-Interim, D1 and D2 forcing, respectively

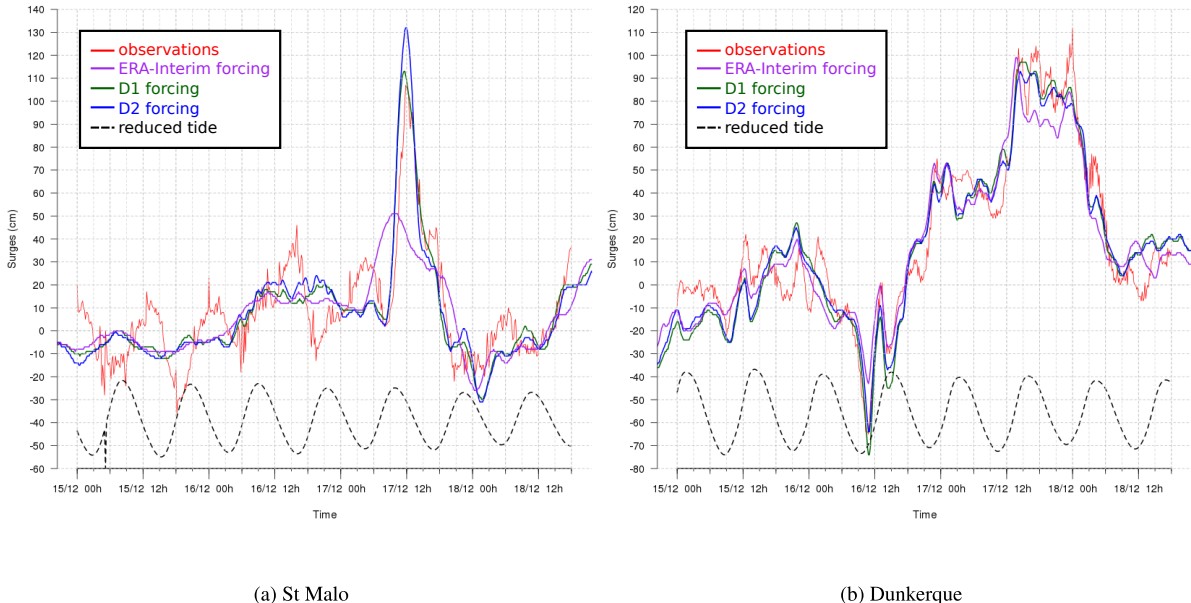

(a) St Malo             (b) Dunkerque

**Figure 9.** Storm surges (cm) at St Malo (a) and Dunkirk (b) from 14 December, 2004, at 15 UTC to 19 December, 2004, at 06 UTC. The measured surge (red line), the reconstructed surge by using the ERA-Interim forcing (purple line), the D1 forcing (green line) and the D2 forcing (blue line) are superimposed. The oscillatory dotted line in the lower part of the graph is used to indicate the time of high and low tides.

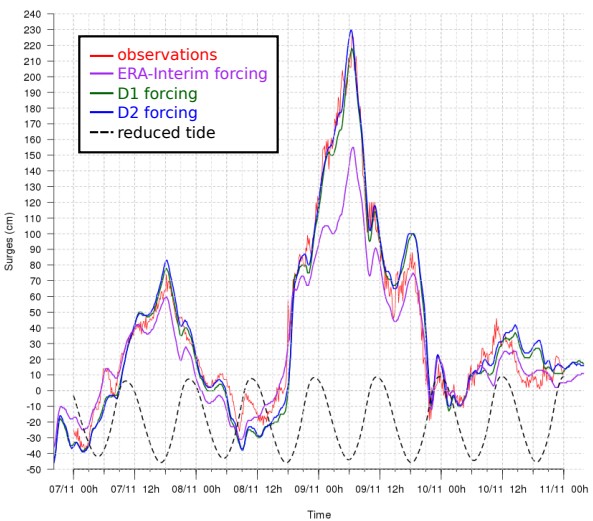

**Figure 10.** Storm surges (cm) at Dunkirk from 7 November, 2007, at 15 UTC to 11 November, 2007, at 06 UTC. The measured surge (red), the reconstructed surge by using the ERA-Interim forcing (purple), the D1 forcing (green) and the D2 forcing (blue) are superimposed. The oscillatory dotted line in the lower part of the graph is used to indicate the time of high and low tides.

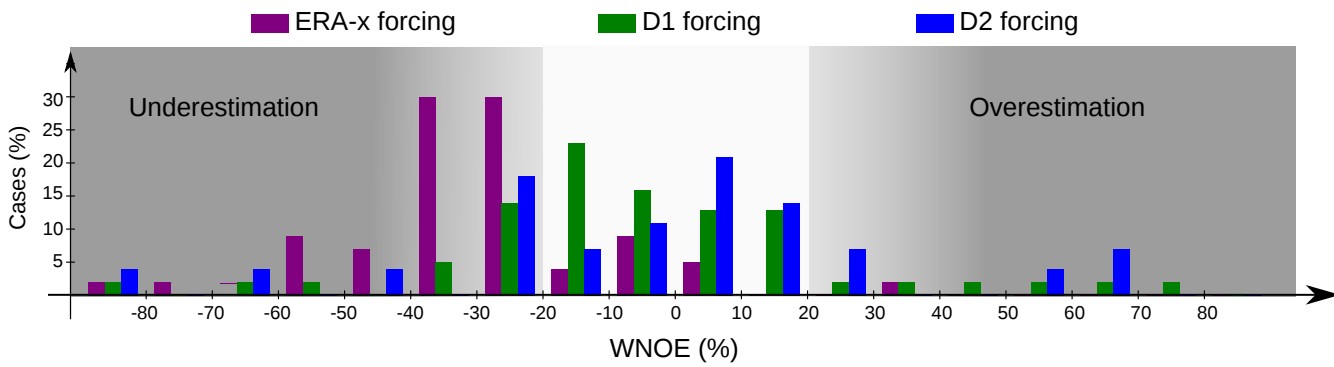

**Figure 11.** The percentage of cases depending of their $WNOE$ range when using ERA-x (purple), D1 (green) or D2 (blue) forcing. All the available observations with a maximum storm surge measurement are taken into account.

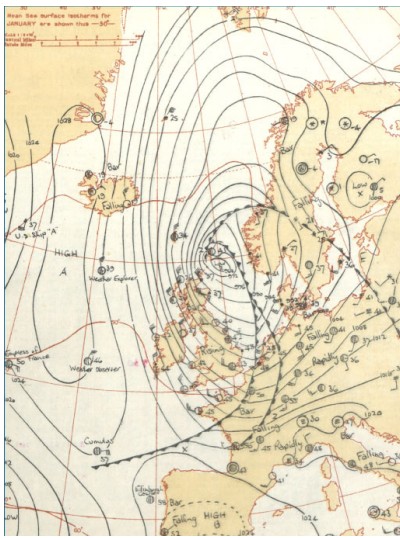

**Figure 12.** Surface pressure chart (hPa) at 06 UTC on 1 February, 1953. From hhtp://www.metoffice.gov.uk

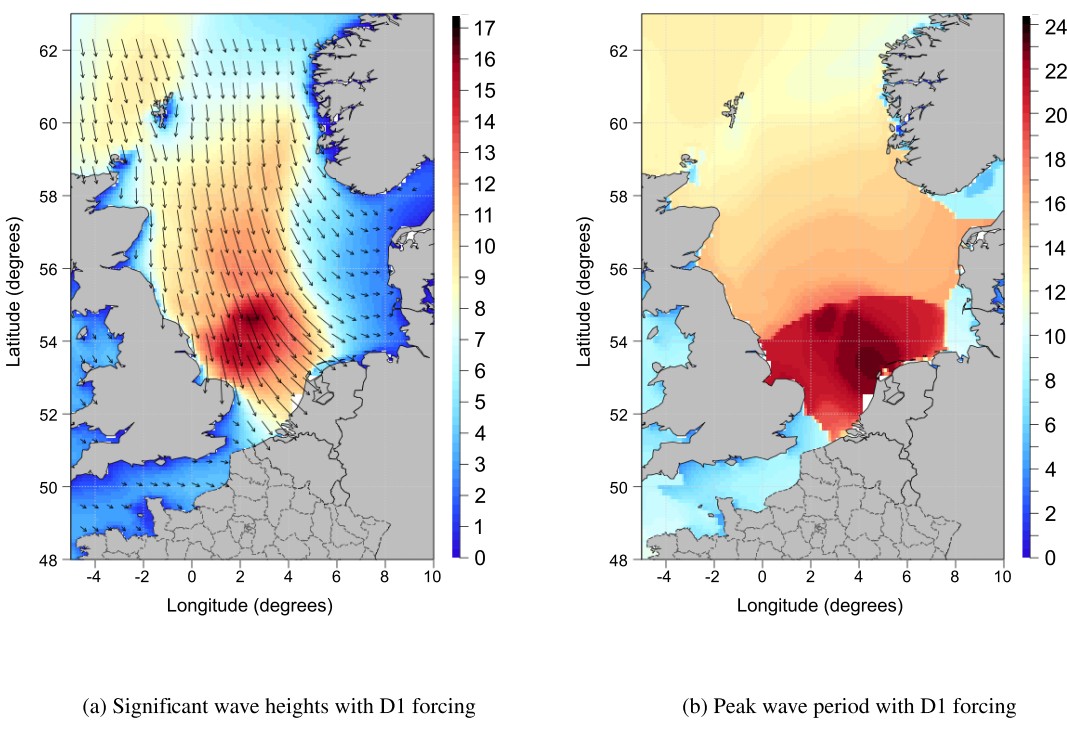

(a) Significant wave heights with D1 forcing      (b) Peak wave period with D1 forcing

**Figure 13.** Significant wave heights (m; a) and peak wave period (s; b) from the wave model MFWAM with D1 winds outputs on the peak of the storm at 00 UTC on 1 February, 1953. Mean Wave Direction is shown with black arrows in (a) when significant wave height are greater than 1.5 m.

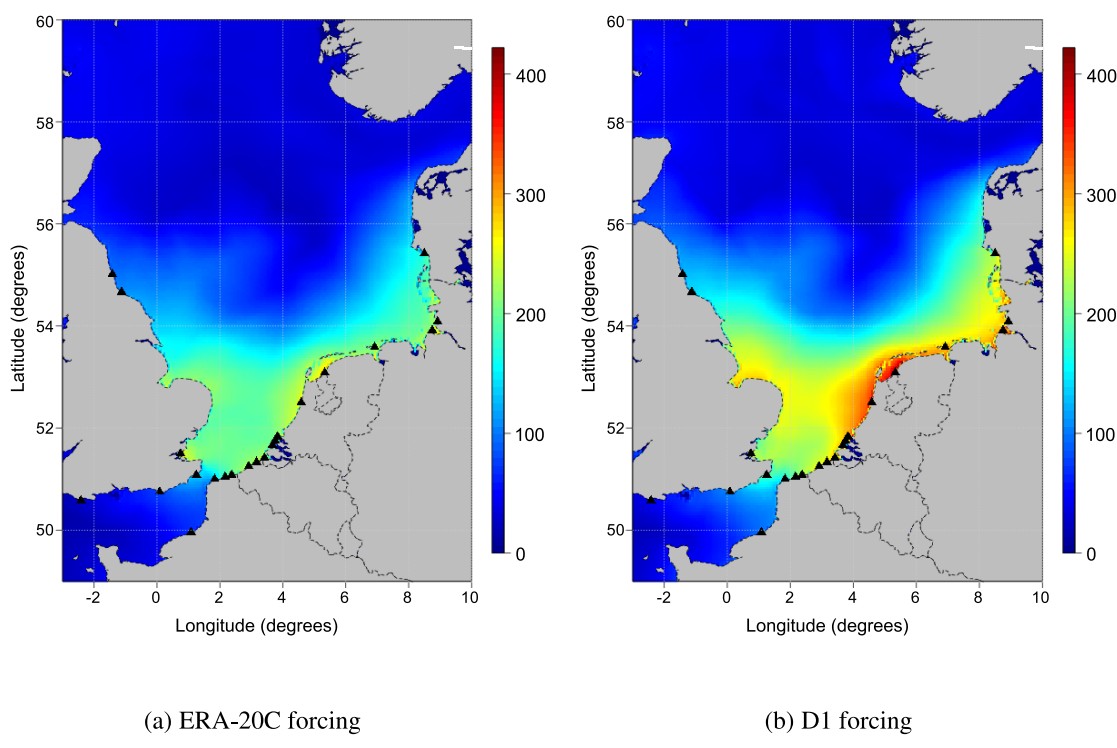

(a) ERA-20C forcing             (b) D1 forcing

**Figure 14.** The highest simulated storm surges (cm) obtained for the period from 30 January to 2 February, 1953, with the ERA-20C forcing (a) and with the D1 forcing (b) along the southern North Sea coast.

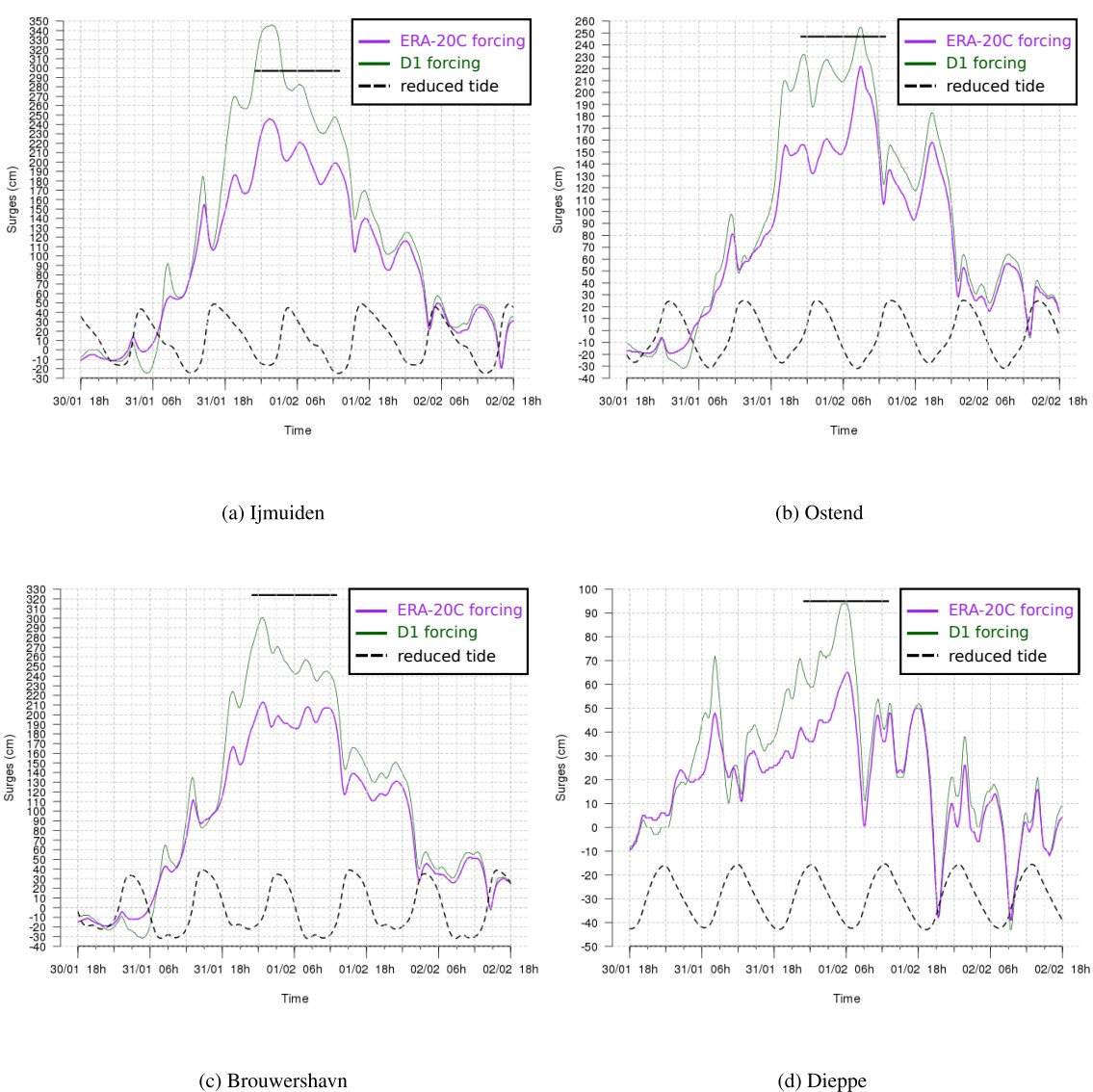

(a) Ijmuiden

(b) Ostend

(c) Brouwershavn

(d) Dieppe

**Figure 15.** The storm surges (cm) at Ijmuiden, Netherland (a), Ostend, Belgium(b), Brouwershavn, Netherlands (c) and Dieppe, France (d) from 18 UTC on 30 January to 18 UTC on 2 February, 1953. Two surges are represented: those resulting from ERA-20C forcing (purple) and from the D1 outputs (green). The maximum observed storm surge is added (horizontal plain black line). The tide level is indicated by the dashed black line (at a reduced scale).