# Peer review of "On the improvement of waves and storm surge hindcasts by downscaled atmospheric forcing: Application to historical storms"

_Natural Hazards and Earth System Sciences, 2017_

## Referee Comment (RC1) · Anonymous Referee #1 · 8 Jun 2017

Review of the Natural and Hazards Earth System Sciences - Discussions: "On the improvement of waves and storm surge hindcasts by downscaled atmospheric forcing: Application to historical storms" by Bresson et al.

The manuscript presents an interesting study on the quality of wave hindcasts and storm surges as a function of different wind forcings. Despite the usefulness of the study I have some concerns regarding the methodology, and serious concerns regarding: knowledge of basic concepts, references completeness, the way results are presented, and, most of all, the quality of the text. There is only so much one can correct, hence the authors should revise the text beyond my corrections. My advice is to move

the manuscript to a second iteration of reviewing, pending on the (major) corrections and alerts I raise below.

P1, L1: What is FP7 IncREO? Please define all acronyms properly. P1, L7: Add "directly" after "wind". P1, L8: What is $D blending? P1, L11: Replace "soil" with "soils". P1, L13: Add "storm" after "events". P1, L17: beset? P1, L18: Intense storms have by definition high winds speeds, no? What are "high winds"? High wind speeds might be better. What are "powerful swells"? All this sentence is full of inaccuracies and lose use of non-scientific and inaccurate terms. Please revise. P2, L3: Global change of what? P2, L17: Replace "model" with "models". P2, L18: What do you mean with "not fully resolved"? P2, L21: Sentence starting with "The technique..." is confusing. Please revise. P2, L25: Add "subsequent" before "dynamical". Replace "fully" with "better" (are you sure that you can "fully" resolve it?). P3, L3-5: All paragraph is confusing. Revise. P3, L6: Which reanalyses? You start talking about reanalysis somehow "out of the blue". How do you know they don't resolve windstorms? This statement is speculative, inaccurate, and not backed up by previous studies (references). Reanalyses (agains tehre are lots of reanalysis, s at this stage we don't know about which one(s) you are talking about) might not resolve properly or accurately extreme wind events, but your statement is not correct. Also, the tense of the verb (here and in several other parts of the text) is not correct. Revise. [You tend to make small paragraphs, sometimes with only one sentence, which is not grammatically correct, and makes the text harder to read. Revise the all text body on this aspect.] P3, L8-10: All paragraph is confusing. Please be clearer. You don't interpolate the reanalysis itself, but its parameters or output. P3, L13: small scales of what? What are "small scales"? [Section 2.2: The way the ECMWF ERA-40, ERA-Interim, and ERA-20C is far from correct. Revise the whole section with accurate statements and use of concepts, backed up by the proper references.] P3, L18-19: Wrong definition of reanalysis. Revise. P3, L19: Define ECMWF. What is ERA? (ERA stands for "European reanalysis"). Define acronyms properly. P4, L12-13: What do you mean with the sentence starting with "Different...". Revise. P4, L16: add "is" before "based". P4, L30: What is MFWAM.

Define. (It is "MFWAM wave model", and not "wave MFWAM model"). Is it correct to call the Meteo France version of WAM as the MFWAM model? What is the Meteo France version of WAM? At the ECMWF the use of ECWAM (the ECMWF WAM version) can be backed up by references that explain its differences to the cycle 4 WAM. Provide the same time of references to sustain the use of the MFWAM acronym and what is stands for. P5, L1-2: Why is the SWAM wave model used (and not implemented) to "investigate extreme events"? (Extreme events of what, by the way?) P5, L16: Add "the" before "Bulgarian". P5, L18: Add references after "lakes". P5, L23: What is BUL? P6, L6: What is NEA? P6, L25-32: The explanation provided in this paragraph falls short. Revise, having in mind clarity and completeness. P7, L1-3: Regarding the sentence stating with "Nevertheless...": where do you see this in your findings? How can you back this statement? Stronger and deeper cyclones where? P7, L5-6: Sentence starting with "NWP...": confusing and speculative. Revise. What is NWP? What is NWP with downscaling? Shouldn't it be something like "The downscaling of atmospheric parameters, like the wind field, using NWP models provide...". P7, L7: From here on you start using the acronym ERA for all reanalysis. That is not correct. You should explain and mention the exact reanalysis you are using. Comparing D1 with any of the three reanalysis means self-correlation, hence that is not correct. P7, L12: Replace "are" with "can be"; replace "all the available data, such as" with ", for example,". P7, L13: Conceptually "evaluate" and "validate" are not the same, and you use these two words for the same task, which is to evaluate the model(s) output metrics by comparison with observations. You do not validate models, hence evaluate its performance. Correct here and in the remainder of the manuscript. P7, L14: Replace "show" with "have shown". P5, L15: Replace "hindcasts" with "the wave hindcast". P7, L18: What is the depth of the ADCP measurements? P7, L25: "model"? Which model? You don't "evaluate models" but their output. Erase "case". P7, L28: Add "the modelled" before "significant". P8, L12: Replace collected with "used"; replace results" with "modelled wave heights". P8, L16: Replace "the good", with "a good". P8, L23: Avoid the use of expressions like "perfectly", since they do noy fit in academic writing standards. P8,

L32: Replace "considered" with "chosen". P9, L4: Replace "winds" with "wind speeds"; add "during" after "as". P9, L5: Add "the" before "December". (I am afraid you do this mistake several times; please check the whole text.) P9, L11: add "results" after "surge". P9, L14: They? Who ate "They"? Revise this sentence. P9, L18: reported or measured? [The whole section 4.3 is very confusing. Revise.] P10, L3: replace "features" with "speed events". P10, L7: which reanalysis? [The conclusions section is also confusing, and way too simplistic/simplified. Revise carefully in line with the findings you convey in the manuscript.]

---

## Referee Comment (RC2) · Anonymous Referee #2 · 11 Jun 2017

The authors have attempted to present the results of an atmospheric downscaling with application to wave and storm surge hindcasting. It is indeed a very laudable project as it is well known that global atmospheric reanalyses currently available struggle to provide a good estimation of storm wind intensity, and hence waves and storm surge conditions needed to evaluate future hazards. This manuscript however reads more like a technical report than a paper suitable for a journal. It is my opinion that upon some restructuring, a clearer separation of the narrative (selected cases) from the statistical analysis it should become suitable for publication. Two separate areas were analysed (the French Coast up to the southern North Sea and the Bulgarian coast). Nevertheless, the manuscript currently feels like two separate papers written by two

separate teams. It will greatly benefit it this duality was removed as much as possible and one common narrative was presented. Specific comments: Introduction: The downscaling of ECMWF reanalysis has been done before, For instance the Norwegian NORA-10 based on ERA-40: Reistad, M., Ø. Breivik, H. Haakenstad, O. J. Aarnes, B. R. Furevik, and J. Bidlot (2011), A high-resolution hindcast of wind and waves for the North Sea, the Norwegian Sea, and the Barents Sea, J. Geophys. Res., 116, C05019, doi:10.1029/2010JC006402. Page 2, line 18: "extreme convective systems" . This seems to imply that only convective systems have very strong winds. Deep winter lows will produce very high winds but they are not necessarily what would be described as convective systems Page 3, section 2.1: Can you be more specific on the interpolation method. ECMWF uses a spectral representation of their atmospheric fields with grid point representation for the surface fields (and a few others). What was done exactly? What about the vertical? Later it is mentioned that the 6 first hours of each forecast were discarded to avoid spin-up effect. How does this 6 hour window relate to the interpolation method? Why not 3 hours, instead? Does it have any impact on the results? Page 4, line 3: both ERA-40 and ERA-Interim were reanalysis for land and waves as well atmosphere, just as ERA-20C. Section 3.3: What is the justification of not coupling the storm surge model with the wave model. There are ample evidence that it is beneficial to both surge and waves hindcast. See for instance Bertin et al. (2015). Xavier Bertin, Kai Li, Aron Roland, Jean-Raymond Bidlot. 2015: The contribution of short-waves in storm surges: Two case studies in the Bay of Biscay. Continental Shelf Research 96, 1-15.

Section 4: It is not clear which is time discretisation of the different forcing. ERA-Interim analysis data are 6-hourly and could be supplemented with 3-hourly forecasts to yield 3-hourly forcing. The down-scaled D1 and D2 fields, I assume are hourly. Consolidate and summarise the statistical analysis by avoiding to show statistics on very short time series but rather on the full sample and use instead the few selected cases as qualitative examples on the type of differences that was obtained.

---

## Author Comment (AC1) · 10 Jul 2017

We thank this reviewer for taking time to review our paper and for making constructive comments. We are currently working on a revised version of the manuscript including all the corrections proposed by this reviewer.

---

## Author Comment (AC2) · 10 Jul 2017

We thank this reviewer for taking time to review our paper and for making constructive comments. We are currently working on a revised version of the manuscript including a reorganization of the text and taking into account all the corrections proposed by this reviewer.

---

## Author Response (AR1)

Émilie Bresson

Université du Québec en Abitibi-Témiscamingue
550 Rue Sherbrooke W.
West Tower, 19th floor
H3A 1B9 Montreal (QC)
Canada

Montreal, September 29th 2017

**Response to reviewers' comments**

Dear Editor,

First, we want to warmly thank the two reviewers for their helpful and constructive comments and suggestions. Their involvement has enabled us to consistently improve the quality and, hopefully, the readability of our manuscript.

The main text has been substantially restructured to make it more flowing and avoid the duality between both regions of interest (France and Bulgaria). We now focus more wave and storm surge hindcasts evaluations. Thus, figures have been rearranged, three tables and one figure have been added. The Section 2 (technical aspects) is now divided in two parts: downscaling techniques explanations followed by wave and storm surge models descriptions. Results (Section 3) presents first an example for downscaled meteorological fields, second the wave hindcast study, then the storm surge hindcast and finally the early 20th century cases. For all the results subsections, a global analysis is provided before the presentation of some examples. The introduction and conclusions parts have been also restructured and completed to better highlight the importance of the downscaling methods to provide better atmospheric forcings for wave and storm surge models that represents the core of the paper.

Here is a detailed answer to all comments from the two reviewers. We hope the new version of our manuscript is of publishable standard for your journal. We remain at your disposal and will gratefully receive any additional comments and suggestions.

On behalf of the co-authors,

Émilie Bresson

**Response to Anonymous reviewer #1**

The manuscript presents an interesting study on the quality of wave hindcasts and storm surges as a function of different wind forcings. Despite the usefulness of the study I have some concerns regarding the methodology, and serious concerns regarding: knowledge of basic concepts, references completeness, the way results are presented, and, most of all, the quality of the text. There is only so much one can correct, hence the authors should revise the text beyond my corrections. My advice is to move to a second iteration of reviewing, pending on the (major) corrections and alerts I raise below.

Thanks a lot for your interest in our study. We hope that we adequately addressed your suggestions in the new version of our manuscript, and that you consider our paper worthy of publication.

P1, L1: What is FP7 IncREO? Please define all acronyms properly.

FP7 IncREO is the "Seventh Research Framework Programme Increasing Resilience through Earth Observation". The modification has been made in the revised manuscript.

P1, L7: Add "directly" after "wind".

Done.

P1, L8: What is 4D blending?

4D blending stands for 4-dimensional blending.
The modification has been made in the revised manuscript.

P1, L11: Replace "soil" with "soils".

Done.

P1, L13: Add "storm" after "events".

Done.

P1, L17: beset?

The right word is "hit". The modification has been made in the revised manuscript.

P1, L18: Intense storms have by definition high winds speeds, no? What are "high winds"? High wind speeds might be better. What are "powerful swells"? All this sentence is full of inaccuracies and lose use of non-scientific and inaccurate terms. Please revise.

We modified the phrasing of "high winds" in "high wind speeds" according your comments, as well as "Powerful swells" modify for "strong swells". Furthermore a quantitative example of "high wind speeds"

and "powerful swells" is presented in the Introduction with the description of the Xynthia storm to present some representative values of what we call "strong wind speeds" and "strong swells".

P2, L3: Global change of what?

It is global change of climate. The modification has been made in the revised manuscript.

P2, L17: Replace "model" with "models".

Done.

P2, L18: What do you mean with "not fully resolved"?

"Not fully resolved" was removed in the revised text. The point here is the fact that with a coarse resolution the reanalyses do not represent the wind characteristics providing from the small-scale processes.

P2, L21: Sentence starting with "The technique. . ." is confusing. Please revise.

This sentence is revised with a more detailed explanation:
*Although we can use the finer reanalysis as initial conditions for a given event it turns out that mesoscale processes related to the formation of strong winds such as sting jets (Hewson and Neu, 2015) are absent even in ERA-Interim, the reanalysis with the highest resolution. As described in Reistad et al. (2011); Li et al. (2016), a dynamical downscaling can be applied on these reanalyses using high resolution numerical model to better resolve the horizontal scales involved together in the mid-latitude cyclone development processes and interaction with fine resolution coastal topography.*

P2, L25: Add "subsequent" before "dynamical". Replace "fully" with "better" (are you sure that you can "fully" resolve it?).

Done.

P3, L3-5: All paragraph is confusing. Revise.

The paragraph is modified:
*Mean sea level pressure and surface wind are usually needed as atmospheric forcing to forecast wave and storm surges. In the present study, these two climate variables are obtained through reanalyses from a numerical weather prediction system conducting data assimilation to encompass past observation datasets. A dynamical downscaling is applied on global atmospheric reanalyses, since their resolution is too coarse to deliver accurate information for hindcast.*

P3, L6: Which reanalyses? You start talking about reanalysis somehow "out of the blue". How do you know they don't resolve windstorms? This statement is speculative, inaccurate, and not backed up by previous studies (references). Reanalyses (again there are lots of reanalysis, s at this stage we don't know about which one(s) you are talking about) might not resolve properly or accurately extreme wind events, but your

statement is not correct. Also, the tense of the verb (here and in several other parts of the text) is not correct. Revise.

All this part is modified with a restructuration of the text for the technical part. The Sections 2.1, 2.2 and 3.1 are mixed to give a clearer information about reanalyses, atmsopheric models selected and downscaling methods (Section 2.1).

You tend to make small paragraphs, sometimes with only one sentence, which is not grammatically correct, and makes the text harder to read. Revise the all text body on this aspect.

The whole text was revised in this way.

P3, L8-10: All paragraph is confusing. Please be clearer. You don't interpolate the reanalysis itself, but its parameters or output.

The paragraph was revised and made clearer. The whole downscaling methods section is modified.

P3, L13: small scales of what? What are "small scales"?

The phrasing of "small scale" referred to scale beyond the truncature of the reanalysis.

Section 2.2: The way the ECMWF ERA-40, ERA-Interim, and ERA-20C is far from correct. Revise the whole section with accurate statements and use of concepts, backed up by the proper references.

Section 2.2 is now merge with Sections 2.1 and 3.1 and becomes Section 2.1. The new Section 2.1 includes a corrected description of the three ECMWF reanalyses used in this study (Tab. 2), of the atmospheric models selected and the two downscaling methods.

Table 2. Characteristics of ERA-20C, ERA-40 and ERA-Interim reanalyses. 4(3)D-Var: 4(3)-dimensional variational analysis; VarBC: Variational Bias Correction of surface pressure observations.

| | ERA-20C | ERA-40 | ERA-Interim |
|---|---|---|---|
| Time period | 1900 – 2010 | 1957 – 2002 | 1979 – present |
| IFS version | Cy38r1 | Cy23r4 | Cy31r2 |
| Data assimilation system | 24-hour 4D-Var; VarBC | 6-hour 3D-Var | 12-hour 4D-Var; VarBC |
| Spectral resolution | T159 (~ 125 km) | T159 (~ 125 km) | T255 (~ 80 km) |
| Number of vertical levels | 91 | 60 | 60 |
| Vertical scale (from the surface up to) | 0.01 hPa (~ 80 km) | 0.1 hPa (~ 64 km) | 0.1 hPa (~ 64 km) |
| Pressure levels | 37 | 23 | 37 |
| Reference | Poli et al. (2013) | Uppala et al. (2005) | Dee et al. (2011) |

P3, L18-19: Wrong definition of reanalysis. Revise.

The definition of a reanalysis is modified by this sentence:
*A global atmospheric reanalysis is built using a data assimilation system and historical observations spanning an extended period.*

P3, L19: Define ECMWF. What is ERA? (ERA stands for "European reanalysis"). Define acronyms properly.

ECMWF is "European Centre for Medium-Range Weather Forecasts". ERA is "European reanalysis". The modifications have been made in the revised manuscript.

P4, L12-13: What do you mean with the sentence starting with "Different. . .". Revise.

Three coastlines, in two countries, are studied. Two options could have fit this study: working with the same atmospheric, wave and storm-surge models for the two countries, or select the more efficient model for each country and with similar technical characteristics. We considered the second approach as our goal is to present efficiency of the downscaling methods in different frames.

P4, L16: add "is" before "based".

Done.

P4, L30: What is MFWAM (It is "MFWAM wave model", and not "wave MFWAM model"). Is it correct to call the Meteo France version of WAM as the MFWAM model? What is the Meteo France version of WAM? At the ECMWF the use of ECWAM (the ECMWF WAM version) can be backed up by references that explain its differences to the cycle 4 WAM. Provide the same time of references to sustain the use of the MFWAM acronym and what is stands for.

The Meteo-France WAve Model (MFWAM) is a third-generation model of the operational wave forecasting system of Météo-France. This model is based on the IFS-CY36R4 of the European wave model (ECWAM) with modified source terms for the dissipation by wave breaking and the air friction dedicated to swell damping as described in Ardhuin et al. (2010). This point is corrected in Section 2.2.1.

P5, L1-2: Why is the SWAM wave model used (and not implemented) to "investigate extreme events"? (Extreme events of what, by the way?)

In a general way, the SWAM model is the operational wave model for Bulgaria. We decided to use this model for studying extreme events of high wave, even if it totally fit with the whole types of events.
The "extreme events" referred to the studied cases of high wave cases.

P5, L16: Add "the" before "Bulgarian".

Done.

P5, L18: Add references after "lakes".

Done.

P5, L23: What is BUL?

BUL stands for the Bulgarian domain.

P6, L6: What is NEA?

NEA stands for North East Atlantic Ocean. The text is modified.

P6, L25-32: The explanation provided in this paragraph falls short. Revise, having in mind clarity and completeness.

P7, L1-3: Regarding the sentence stating with "Nevertheless. . .": where do you see this in your findings? How can you back this statement? Stronger and deeper cyclones where?

The two last comments are about the lack of clarity of the subsection. Actually, the main purpose of the paper is on wave and storm surge hindcast. The present subsection is just an example of how the atmospheric component of the downscaling methods behave in the case of a very strong and relatively small scale cyclone development although it is not associated with strong waves and surge. The section is revised along that lines. The last sentences are suppressed.

P7, L5-6: Sentence starting with "NWP. . .": confusing and speculative. Revise. What is NWP? What is NWP with downscaling? Shouldn't it be something like "The downscaling of atmospheric parameters, like the wind field, using NWP models provide. . .".

A NWP system is a Numerical Weather Prediction system. In the restructuration of the text, this part has been suppressed.

P7, L7: From here on you start using the acronym ERA for all reanalysis. That is not correct. You should explain and mention the exact reanalysis you are using. Comparing D1 with any of the three reanalysis means self-correlation, hence that is not correct.

The acronym ERA was used to lighten the text, the complete name of the reanalysis considered is added. The detail of each ERA-storm couple is presented in the Tab. 1.
We do not compare reanalysis fields with downscaled fields, we just compare the departure between D1 and the downscaled observations and departure between the reanalysis fields and observations.

P7, L12: Replace "are" with "can be"; replace "all the available data, such as" with ", for example,".

Done.

P7, L13: Conceptually "evaluate" and "validate" are not the same, and you use these two words for the same task, which is to evaluate the model(s) output metrics by comparison with observations. You do not validate models, hence evaluate its performance. Correct here and in the remainder of the manuscript.

"Validate" is modified by "evaluate" in the whole text.

P7, L14: Replace "show" with "have shown".

Done.

P5, L15: Replace "hindcasts" with "the wave hindcast".

Done.

P5, L18: What is the depth of the ADCP measurements?

The ADCP is located at 20 m depth.

P7, L25: "model"? Which model? You don't "evaluate models" but their output. Erase "case".

The word "case" is erased.

P7, L28: Add "the modelled" before "significant".

Done.

P8, L12: Replace collected with "used"; replace results" with "modelled wave heights".

Done.

P8, L16: Replace "the good", with "a good".

Done.

P8, L23: Avoid the use of expressions like "perfectly", since they do not fit in academic writing standards.

The modification has been done in the revised manuscript.

P8, L32: Replace "considered" with "chosen".

Done.

P9, L4: Replace "winds" with "wind speeds"; add "during" after "as".

Done.

P9, L5: Add "the" before "December". (I am afraid you do this mistake several times; please check the whole text.)

Done.

P9, L11: add "results" after "surge".

Done.

P9, L14: They? Who ate "They"? Revise this sentence.

"They" stands for the ERA-20C reanalyses. The sentence is modified.

P9, L18: reported or measured?

The right word is "measured". The sentence is modified.

The whole section 4.3 is very confusing. Revise.

The Section 4.3 (now Section 3.4) is revised for an easier reading.

P10, L3: replace "features" with "speed events".

Done.

P10, L7: which reanalysis?

The term reanalyses encompassed the three selected reanlayses. The sentence is modified.

The conclusions section is also confusing, and way too simplistic/simplified. Revise carefully in line with the findings you convey in the manuscript.

The conclusions are revised and completed to better resume study findings and future work possibilities.

**Response to Anonymous reviewer #2**

The authors have attempted to present the results of an atmospheric downscaling with application to wave and storm surge hindcasting. It is indeed a very laudable project as it is well known that global atmospheric reanalyses currently available struggle to provide a good estimation of storm wind intensity, and hence waves and storm surge conditions needed to evaluate future hazards. This manuscript however reads more like a technical report than a paper suitable for a journal. It is my opinion that upon some restructuring, a clearer separation of the narrative (selected cases) from the statistical analysis it should become suitable for publication. Two separate areas were analysed (the French Coast up to the southern North Sea and the Bulgarian coast). Nevertheless, the manuscript currently feels like two separate papers written by two separate teams. It will greatly benefit it this duality was removed as much as possible and one common narrative was presented.

Thanks a lot for your strong interest in our study. We hope that the restructuring of the text responds to your suggestions as to the technicality and duality present in this paper.

The article structure has been revised in order to avoid the technical aspect and the feeling of study in two distinct parts. As a consequence, the Introduction was restructured. The Section 2 (technical aspects) is them divided in two parts: downscaling techniques explanations and wave and storm surge models descriptions. Section 3 presents the results with evaluation of first the meteorological fields, second the wave hindcast study, then the storm surge hindcast and finally the early 20th century cases. For all the results subsections, a global analysis is provided before the presentation of some examples.

**Specific comments:**

Introduction:

The downscaling of ECMWF reanalysis has been done before, For instance the Norwegian NORA-10 based on ERA-40: Reistad, M., Ø. Breivik, H. Haakenstad, O. J. Aarnes, B. R. Furevik, and J. Bidlot (2011), A high-resolution hindcast of wind and waves for the North Sea, the Norwegian Sea, and the Barents Sea, J. Geophys. Res., 116, C05019, doi:10.1029/2010JC006402.

To acknowledge the contribution by Reistad, et al. (2011), we now quote it in the introduction. Our study is a step forward since we considered three different reanalyses to encompass a large period and different areas.

Page 2, line 18: "extreme convective systems" . This seems to imply that only convective systems have very strong winds. Deep winter lows will produce very high winds but they are not necessarily what would be described as convective systems

The sentence has been modified in the restructuration of the Introduction.

Page 3, section 2.1: Can you be more specific on the interpolation method. ECMWF uses a spectral representation of their atmospheric fields with grid point representation for the surface fields (and a few others). What was done exactly? What about the vertical? Later it is mentioned that the 6 first hours of each forecast were discarded to avoid spin-up effect. How does this 6 hour window relate to the interpolation method? Why not 3 hours, instead? Does it have any impact on the results?

We agree that the lack of a technical description of the global model initialization is a major caveat of the paper. We therefore suggest to add the following sentences:

*The upper-air initialization step is using the spectral coefficients of ERA reanalyses. Then we apply the Schmidt transform which is well defined in spectral space to project the fields into the ARPEGE stretched grid. The land-surface initialization is not straightforward since many differences of land-surface parametrizations and physiographic databases between the two land-surface schemes can be found. For instance, the Tiled ECMWF Scheme for Surface Exchanges over Land (TESSEL) scheme of ERA uses four soil layers with fixed thicknesses, each layer having its own water content. The land-surface scheme of ARPEGE uses only two layers in our experiments, the top layer with a fixed size of 1 cm and the second layer overlaps the first one and has a variable depth. For a given grid point soil types are very different in the two land-surface schemes. Therefore, using the raw land-surface datasets from ERA as initial conditions would be troublesome since the water saturation fraction depends on the soil type. Thus, we interpolate the surface fields so as to preserve as much as possible the surface heat and momentum fluxes (see Boisserie et al., 2016 for a thorough description of how we proceed). The procedure is based on the conservation of the Soil Wetness Index (a relevant indicator for soil water availability) during the interpolation process since soil water availability is supposed to regulate the partition of latent and heat fluxes, which, in turn, influence energy and water exchanges between the atmosphere and the land-surface.*
[Boisserie, M., Decharme, B., Descamps, L. and Arbogast, P. (2016), Land surface initialization strategy for a global reforecast dataset. Q.J.R. Meteorol. Soc., 142: 880–888. doi:10.1002/qj.2688]

Two points explained this spin-up effect. First dynamically the spectrum needs time to fill up and second, a certain period is necessary to reach physical coherence between surface and fluxes. This spin-up time is about some hours. The 6-h spin-up was selected for more convenience with the 6-hourly ERA reanalyses.

Page 4, line 3: both ERA-40 and ERA-Interim were reanalysis for land and waves as well atmosphere, just as ERA-20C.

The sentence is modified to provide a cleared message.

Section 3.3: What is the justification of not coupling the storm surge model with the wave model? There are ample evidence that it is beneficial to both surge and waves hindcast. See for instance Bertin et al. (2015). Xavier Bertin, Kai Li, Aron Roland, Jean-Raymond Bidlot. 2015: The contribution of short-waves in storm surges: Two case studies in the Bay of Biscay. Continental Shelf Research 96, 1-15.

We thank you for this suggestion and we add this in the future work in the conclusions section. Indeed, results could have been better applying a coupling between wave and storm-surge models. Even if, our objective was to stay close the operational conditions using wave and storm surge models used in the French and Bulgarian centers.

Section 4: It is not clear which is time discretisation of the different forcing. ERA-Interim analysis data are 6-hourly and could be supplemented with 3-hourly forecasts to yield 3-hourly forcing. The down-scaled D1 and D2 fields, I assume are hourly. Consolidate and summarise the statistical analysis by avoiding to show statistics on very short time series but rather on the full sample and use instead the few selected cases as qualitative examples on the type of differences that was obtained.

ERA fields are 6-hourly. Forcing are introduced in the wave and storm surge models every 3 hours when using ERA reanalyses without downscaling and every hour when using atmospheric fields after D1 or D2 method.

The section 3 (Results) is revised in this way. A new figure according to your suggestion is added (Fig. 9). It gives an overall idea for the cases where the observations are available.

For other verification results, it not possible to have a large ensemble: for example, there is no altimeter data for every studied storm.

[Figure]

**Figure 9.** Percentage of cases depending of their $WNOE$ range when using ERA (purple), D1 (green) or D2 (blue) forcing. All the available observations with a maximum storm surge measurement are taken into account.

**Table 6.** Portion of cases (%) with $\|WNOE\| < 20\%$ for each coast (ATL: Atlantic; MED: Mediterranean Sea; BUL: Bulgarian; common cases: cases using D1 and D2 forcing).

|              | ERA | D1  | D2  |
|--------------|-----|-----|-----|
| ATL          | 21  | 63  | 80  |
| MED          | 0   | 54  | 38  |
| BUL          | 33  | 100 | –   |
| Common cases | 18  | 64  | 61  |

**Table 7.** Number of observations used for calculations of $WNOE$ for each region and each forcing.

|     | ERA | D1 | D2 |
|-----|-----|----|----|
| ATL | 34  | 34 | 15 |
| MED | 13  | 13 | 13 |
| BUL | 9   | 9  | 0  |

---

## Author Response (AR2)

Émilie Bresson

Université du Québec en Abitibi-Témiscamingue 550 Rue Sherbrooke W. West Tower, 19th floor H3A 1B9 Montreal (QC) Canada

Montreal, December 14th 2017

**Response to reviewer' comments**

Dear Editor,

First, we want to warmly thank the reviewer for her/his very constructive comments and suggestions. Her/his involvement have enabled us to consistently improve the clarity and, hopefully, the quality of our manuscript.

We conducted five main actions to improve our manuscript:

- The structure of the introduction was modified for more fluency. The introduction now follows a clearer thread: context, motivation, purpose of the study. We hope we provided more clarity for the aim of the study.
- Our study has two goals: first, comparing surge and wave reconstruction against observations while using either ECMWF reanalysis data or D1 data as forcing; then examining if using D2 has an added-value compared to using D1 data as forcing. In the Results section, the text in the sections 3.2 and 3.3 has been reorganized in order to answer these two questions, one after the other.
- The two downscaling methods (D1 and D2) descriptions are now more detailed and supported by the modified Fig. 1. The figure highlights the similarities and differences between D1 and D2. We also explained more explicitly our choice of testing D2 method only on 10 cases instead of the whole 30 historical storms studied.
- The term "ERA" now only referred to the acronym of ECMWF Re-Analysis. In the other cases, the confusing term of "ERA" was replaced by the name of the ECMWF reanalysis (i.e. ERA-20C, ERA-40 or ERA-Interim) when only one type of ECMWF reanalysis is considered or by "ERA-x" when two or more types of ECMWF reanalysis are used.
- At the end of our revision, the manuscript has been reviewed by a colleague with English as native language who paid a particular attention to grammar, structure and fluidity of the text.

Here is a detailed answer to all comments from the reviewer. We hope the new version of our manuscript will please the reviewer and you, and is of publishable standard for your journal. We remain at your disposal and will gratefully receive any additional comments and suggestions.

Best regards, On behalf of the co-authors, Émilie Bresson

**Response to Anonymous reviewer #1**

Unfortunately the concerns regarding the methodology, and the quality of the text are still present. The text still feels more like a technical report, and still needs improvement. I raise some questions below, but there are much more to be addresses. The text is also full of grammatical errors that should have not been passed to a second review iteration. Please revise once more.

We want to thank you for your involvement in the revision for our manuscript. Your comments and suggestions highlight very well the mistakes and the disorganization present in the previous version of the manuscript.

The introduction was reorganized and the aim of our study is now, hopefully, clearer. The analysis of our wave and storm surge reconstructions is now following the two points of our goal.

In order to prevent grammatical errors and other language mistakes, we ask a colleague with English as native language to revise and correct our manuscript.

We hope that the new version of our manuscript would answer all your concerns and that it will offer you the opportunity to evaluate in a more pleasant and fluid way our study.

P1-L5: Add the ECMWF reanalysis acronyms you are using between curly brackets after "reanalyses".

We modified the abstract following this point: the ERA-20C, ERA-40 and ERA-Interim acronyms have been added between curly brackets in the abstract.

P2-L8-9: Are you doing this? "A global atmospheric reanalysis is built using a data assimilation system and historical observations spanning an extended period."

This sentence aims at giving a definition of a global atmospheric reanalysis, but the way we express this point was confusing. It has been replaced in the introduction by: "Several weather forecast centers produce these global atmospheric reanalysis, including the European Centre for Medium-Range Weather Forecasts (ECWMF). The ECMWF Re-Analyses (ERA) include different products that have various date ranges, spatial resolutions and assimilated datasets (Tab. 1; Poli et al., 2013; Uppala et al., 2005; Dee et al., 2011)."

P2-18: ERA is not "European Reanalysis" but "ECMWF Re-Analysis".

The definition of the ERA acronym as "ECMWF Re-Analysis", and not "European Reanalysis", has been corrected in the revised manuscript.

P2-20: The ECMWF reanalyses were not produce using a "coupled climate system, including atmosphere, land surface and ocean". They have been produced by older versions of IFS, the ECMWF operational forecasting coupled model system. By "older" is meant a previous version, compared to the present (at the time) operational version.

The definition of the ECMWF reanalysis has been corrected in the revised manuscript in the section 2.1.

P2-24: ERA-Interim is not "the reanalysis with the highest resolution". There are several other reanalyses with higher resolution, like CFSR, MERA2, JRA-55, etc. ERA-Interim is the highest resolution reanalysis at ECMWF, so your statement is misleading.

**We totally agree with you and we add more details about this point in the introduction: "ERA-Interim, one of the higher-resolution reanalyses available from the ECMWF."**

P3-L2-5: This sentence (first paragraph on page 3) is full of inaccuracies and misleading statements: "Mean sea level pressure and surface wind are needed as atmospheric forcing to forecast wave and storm surges. In the present study, these two variables are obtained through reanalyses built using a given data assimilation system constrained by past observations. A dynamical downscaling is applied on global atmospheric reanalyses since their resolution is too coarse to deliver accurate information for hindcast."

1. Not "obtained through" but "obtained from".

**Done.**

2. The "downscaling" is not "applied" but a "dynamical downscaling of the global reanalyses is produced, since the global reanalyses are too coarse to force the regional wave and storm surge models".

**The sentence has been corrected in the revised manuscript.**

P6-L21-22: Stating the goal of the study in the results section is not correct. Move it to the introduction.

Done.

**P8-L19: Which ERA?**

For this storm (November 2007), the ECMWF reanalysis used is ERA-Interim.

In the new version of the manuscript, we explicitly use the name of the ECMWF reanalysis considered if only one type of reanalysis is used (ERA-20C or ERA-40 or ERA-Interim). When more than one type of reanalysis is used, ECMWF reanalysis are named ERA-x.

P1-L1: Replace "analysis" with "reanalysis".

Done.

P16-16: Why coarse resolution generates "strong uncertainties"?; What is a "strong uncertainty"?

The correct words were "significant bias" instead of strong uncertainties. The modification has been made. With this "significant bias", we refer to the mesoscale features associated with mid-latitude cyclone development which are not represented in the coarse resolution reanalysis.

**On the improvement of waves and storm surge hindcasts by downscaled atmospheric forcing: Application to historical storms**

Émilie Bresson1, Philippe Arbogast1, Lotfi Aouf2, Denis Paradis2, Anna Kortcheva3, Andrey Bogatchev3, Vasko Galabov3, Marieta Dimitrova3, Guillaume Morvan2, Patrick Ohl2, Boryana Tsenova3, and Florence Rabier4

[revised manuscript text omitted]
 benefit from improvements in assimilation methods and large increase a large expansion of available data. It turns out that , with observation quantity and quality increased with time. Therefore in increasing over time. In order to mitigate the inhomogeneity of this inhomogeneity in the 20th century reanalysis,
- 10 only observations of surface pressure and surface marine winds are assimilated in the ERA-20C dataset. To In order to provide the best possible atmospheric conditions for wave and storm surge hindcast, the following ERA datasets are downscaled for each stormevent: ERA-20C for cases study before 1957, ERA-40 for 1957–1978 period, the 1957–1978 period, and ERA-Interim for storms occurring in 1979 and after thereafter (Tab. 2). The designator "ERA-x" is used in this manuscript to describe a group of cases where more than one ERA reanalysis product is applied.
- 15 Over both Hereafter, this study focuses on the advantages of downscaling global atmospheric reanalysis for the development of wave and storm surge hindcasts. Over both the French and Bulgarian domains, numerical weather prediction (NWP) models need to assure high enough horizontal and time resolutionrequire high horizontal and temporal resolution, especially for the storm surge model hindcast. For French events, the selected model, ARPEGE (Action de Recherche Petite Echelle Grande Echelle), is the operational global primitive-equation NWP system used at Météo-France and is based on the ARPEGE-IFS
- software developed in collaboration with ECMWF (Tab. 3; Courtier et al., 1991). A stretched grid allows for a finer horizontal resolution over France (around 10 km)with a coarser one at the antipodes (60 km). The version used here has 70 hybrid vertical levels from 17 m to about 70 km height. The Bulgarian events are hindcast thanks to from ALADIN (Aire Limitée, Adaptation dynamique, Développement InterNational) for more consistency as the ARPEGE grid is too coarse over this region. ALADIN model, which is a limited-area model based on the ARPEGE system (Radnóti et al., 1995). Its The model's core characteristics are the same as for ARPEGE.

Two dynamical downscaling methods (D1 and D2) are examined here, hereafter referred to as D1 and D2, where D2 represents an improved version of D1. For D1, the necessary part data from the global fields of ECMWF reanalysis ERA-x are interpolated to the plane model domain both on horizontal and vertical scale for each NWP system, ARPEGE and AL-ADIN. The upper-air initialization step is using uses the spectral coefficients of ERA reanalysesERA-x data. Then we apply

30 the Schmidt transform transformation, which is well defined in spectral space to project the fields into the ARPEGE stretched grid. The land-surface initialization is not straightforwardsince many differences of, since there are many differences between the ERA reanalysis and the NWP models in terms of the applied land-surface parametrizations parameterizations and physio-graphic databasesbetween the two land-surface schemes can be found. For instance, the Tiled ECMWF Scheme for Surface Exchanges over Land (TESSEL) scheme of ERA-ERA-x uses four soil layers with fixed thicknesses, each layer having its own

water content. The land-surface scheme of ARPEGE<del>uses only, however, only uses</del> two layers in our experiments,; the top layer with has a fixed size of 1 -emcm, and the second layer overlaps the first one and has a variable depth. For Furthermore, for a given grid point, soil types are often very different in the two land-surface schemes. Therefore, using the raw land-surface datasets from ERA-ERA-x as initial conditions would be troublesome, since the water saturation fraction depends on the soil

- 5 type. Thus, we interpolate the surface fields so as to preserve as much as possible the ERA-x surface heat and momentum fluxes (Boisserie et al., 2016). The procedure is based on the conservation of the Soil Wetness Index (a relevant indicator for soil water availability) during the interpolation process, since soil water availability is supposed to regulate the partition of latent and sensible heat fluxes, which, in turn, influence energy and water exchanges between the atmosphere and the land-surface. The resulting files are initial conditions for NWP forecasts and (IC-1) for the NWP forecasts (Fig. 1, top). Then, hourly forecasts are
- 10 produced twice a day(, at 00 UTC and at 12 UTC), starting from H+06 to H+18. The first six hours are not taken into account to prevent model spin up, and after H+18, the next forecast time is considered (Fig. 1, top). Forecasts are produced from a week (d-7) before to two days (d+2) after the day (d) that the storm impacted the coastline. The D2 method helps is more complex than D1 (Fig. 1, bottom). The D2 method also uses hourly forecasts produced twice a day, at 00 UTC and at 12 UTC, starting from H+06 to H+18, and the forecast start nine days (d-9) before and continue until two days after (d+2) the the day (d) that
- 15 the storm impacted the coastline. Instead of using independent initial conditions (IC-1) like in D1 for the 00 UTC and 12 UTC forecasts, the initial conditions for D2 (IC-2) include information from the last 6-h forecast (Fig. 1, bottom). Consequently, the D2 method allows us to evaluate the importance of taking into account small wavelengths beyond the reanalysis truncation , that are not considered in D1. Furthermore, after a short period of time (3 hours), non-linearities trigger small seales scale processes which are consistent with the large scale. This small scale small-scale information provided by the 6-hour forecast
- 20 is blended with the large scale information given by the interpolated reanalysis (IC-1) (Fig. 1, bottom). This procedure was cycled 4 times two days before the first 00 UTC forecast used as forcing for the wave and storm surge models. Therefore, the determination of one single initial condition (IC-2) uses 4 reanalysesand is then, in some sense, 4-dimensional. The D2 technique is apply on applied to 10 French events recent French coastal flooding events (Tab. 2). These 10 cases represent a diverse panel of events affecting different coastlines with adequate observational data (satellite altimeters and tide gauges) to
- 25 evaluate the reconstruction of the wave and storm surge observations and to enable a comparison between D1 and D2.

**2.2 Description of wave and storm surge models**

For more consistency, In order to ensure consistency in our case studies, the selected wave and storm surge models used here have share similar general characteristics while being adapted to either, despite being adapted specifically either the French or Bulgarian coasts.

**30 2.2.1 Wave models**

[revised manuscript text omitted]

- 20 friction coefficient is  $1.5 10^{-3}$  over the shelf<del>and  $1.5 \cdot 10^{-5}$  over the liquid bottom. The In addition, the</del> depth of the Black Sea mixed layer is considered as a liquid bottom (given the very stable stratification of the Black Sea waters and the shallowness of the mixed layer depth), and as such, the bottom friction coefficient is defined as  $1.5 \cdot 10^{-5}$  over the liquid bottom. Data about the seasonal variations of the Black Sea mixed layer depth are taken from the study by Kara et al. (2009). Without this <del>specific</del> set-up of a liquid bottom, the depth integrated liquid bottom setup, the depth-integrated models for the Black Sea would fail
- 25 to simulate any surgeeven if very, even if strong, constant winds were used as an are used as input. The bathymetry data for the storm surge model (and the wave model) are obtained by digitalization of were obtained by digitizing proprietary maps provided by the Bulgarian military hydrographic service.

**3** Results**

**3.1 Impact of the two downscaling techniques on a deep cyclone development**

30 The effects of the two downscaling techniques on the reconstruction of intense storms are presented with an example of a deep cyclone development. The Lothar stormwas for the case of the Lothar storm, an extreme cyclogenesis event (occurring a few hours before the Martin storm described further in Sections 3.2 and 3.3) in December 1999. It is the most extreme severe storm in terms of pressure gradient, surface wind winds and displacement velocity to hit France to this day within the

observational record (Wernli et al., 2002; Rivière et al., 2010). This storm did not produce extreme wave and storm surge, and thus it was not selected for hindcasts. Nevertheless, it is interesting to look at the behaviour oh of both downscaling strategies for this case where the particular case due to its uniquely tight horizontal pressure gradientare so strong. Fig. 3 shows the comparison of the downscaling strategies applied to this storm. For this storm, the D1 method slightly improves the ERA-

5 Interim reanalysis fieldsbut, but the D2 downscaling better reproduces the cyclone structure over Northern France than D1 and the ERA reanalysis. Statistics are (Fig. 3). A statistical analysis using the mean, the bias, the root mean square error (RMSE) and the standard deviation error (STD) is performed with the 12 meteorological stations available on in an area encompassing the low pressure system (48° N-50° N; 2° E-4° E). Table 4 confirms that downscaling is noticeably better than This analysis confirms that the use of D1 forcing is an improvement compared to using an ERA-Interim reanalysis in regard with respect to

10 surface observations. Table 4 also highlights the slight improvement of results by the The use of D2 -

The purpose of the paper is to measure to what extent the mesoscale features built by the downscaling techniques beyond the reanalysis truncation have an impact in terms of surge and wave reconstruction slightly improves the reconstruction of the observations ((Table 4)).

**3.2 Wave hindcasts**

- 15 Significant wave heights For the wave reconstruction evaluation, simulated Significant Wave Heights (SWH) hindcasts can be evaluated by using, for example, in-situ observations and are compared against observations from satellite altimeter data and in-situ observations. Several satellites operated while some selected stormsoccurred over the French and Bulgarian coasts during the storms: TOPEX-Poseidon (1992–2005), ENVISAT (20021000ERS2 (199510002012), Jason-1-2011), ENVISAT (2002–2013) and Jason-2 (200810002012) and Jason-1 (20021000). Buoys 2013). In addition, buoys and Acoustic Doppler
- 20 Current Profiler (ACDP) ADCP) in-situ provide SWH informationas well. Caveats remaining in . The limited scope of each of these observations datasets 
[revised manuscript text omitted]
 sate | llite track | Pairs |     | Mean            |     | Biais           |       | RMSE            |      | Scatter Ind     |
|--------------|-------------|-------|-----|-----------------|-----|-----------------|-------|-----------------|------|-----------------|
|              |             |       | Obs | ERA-ERA-Interim | D1  | ERA-ERA-Interim | D1    | ERA-ERA-Interim | D1   | ERA-ERA-Interin |
| 7 Feb. 2012  | 08 UTC      | 44    | 3.9 | 3.5             | 4.1 | -0.43           | 0.21  | 0.60            | 0.37 | 0.15            |
|              | 14 UTC      | 76    | 3.6 | 3.2             | 3.8 | -0.41           | 0.15  | 0.66            | 0.57 | 0.18            |
|              | 20 UTC      | 51    | 6.4 | 5.3             | 6.3 | -1.08           | -0.09 | 1.14            | 0.37 | 0.18            |
| 8 Feb. 2012  | 14 UTC      | 43    | 5.6 | 4.4             | 4.7 | -1.22           | -0.94 | 1.37            | 1.16 | 0.24            |

|     | ERA-ERA-x | D1 | D2 |
|-----|-----------|----|----|
| ATL | 34        | 34 | 15 |
| MED | 13        | 13 | 13 |
| BUL | 9         | 9  | 0  |

Table 6. Portion of cases (%) with ||WNOE|| < 20% for each coast (ATL: Atlantic; MED: Mediterranean Sea; BUL: Bulgarian; commoncases: cases using D1 and D2 forcing)Number of observations used for calculations of WNOE for each region and each forcing.

**Table 7.** Number of observations used for calculations of WNOE for each region and each forcing.Portion of cases (%) with ||WNOE||< 20% for each coast (ATL: Atlantic; MED: Mediterranean Sea; BUL: Bulgarian; common cases: cases using D1 and D2 forcing).</td>

|              | ERA-ERA-x | D1  | D2 |
|--------------|-----------|-----|----|
| ATL          | 21        | 63  | 80 |
| MED          | 0         | 54  | 38 |
| BUL          | 33        | 100 | _  |
| Common cases | 18        | 64  | 61 |